 **eLIFE**

# A clathrin coat assembly role for the muniscin protein central linker revealed by TALEN-mediated gene editing

Perunthottathu K Umasankar[1], Li Ma[1†], James R Thieman[1‡], Anupma Jha[1], Balraj Doray[2], Simon C Watkins[1], Linton M Traub[1*]

[1]Department of Cell Biology, University of Pittsburgh School of Medicine, Pittsburgh, United States; [2]Department of Medicine, Washington University School of Medicine, St. Louis, United States

**Abstract** Clathrin-mediated endocytosis is an evolutionarily ancient membrane transport system regulating cellular receptivity and responsiveness. Plasmalemma clathrin-coated structures range from unitary domed assemblies to expansive planar constructions with internal or flanking invaginated buds. Precisely how these morphologically-distinct coats are formed, and whether all are functionally equivalent for selective cargo internalization is still disputed. We have disrupted the genes encoding a set of early arriving clathrin-coat constituents, FCHO1 and FCHO2, in HeLa cells. Endocytic coats do not disappear in this genetic background; rather clustered planar lattices predominate and endocytosis slows, but does not cease. The central linker of FCHO proteins acts as an allosteric regulator of the prime endocytic adaptor, AP-2. By loading AP-2 onto the plasma membrane, FCHO proteins provide a parallel pathway for AP-2 activation and clathrin-coat fabrication. Further, the steady-state morphology of clathrin-coated structures appears to be a manifestation of the availability of the muniscin linker during lattice polymerization.

*For correspondence: traub@pitt.edu

Present address: †Tsinghua University School of Medicine, Beijing, China; ‡Olympus America, Inc., Center Valley, United States

Competing interests: The authors declare that no competing interests exist.

## Introduction

The precise purpose of clathrin-coated vesicles was correctly deduced 50 years ago from thin-section electron micrographs (*Roth and Porter, 1964*). Noticing the very close apposition of a compacted lumenal protein load and a cytosol-oriented 'bristle' coat on opposite faces of striking membrane invaginations, it was astutely reasoned that the bulbous plasmalemma structures ferry macromolecules into the cell interior in a highly selective manner. While it was soon clear clathrin-mediated endocytosis is a rapid activity (*Brown and Goldstein, 1976*), and is importantly involved in many fundamental cellular processes (*McMahon and Boucrot, 2011*; *Brodsky, 2012*; *Kirchhausen et al., 2014*), the regularity of the static coated structures in micrograph views masks the true underlying molecular dynamics and complexity. It is also challenging to identify the very earliest stages of clathrin coat assembly in thin sections because of the restricted size, ill-defined morphological features, and a lack of prominent membrane curvature. With the recent live-cell clarification of the linked temporal steps that define clathrin-coated structure initiation, growth, invagination and scission (*Kaksonen et al., 2003*; *Taylor et al., 2011*), it is clear that a large number of gene products are involved. The choreographed arrival and departure of these numerous proteins matches well their defined biochemical activities and the morphological progression of the bud (*Taylor et al., 2011*). Yet, what exactly defines a restricted plasma membrane zone that is destined to become a clathrin-coated bud remains elusive. In this regard, the clathrin coats at the cell surface are distinct from many other intracellular cargo sorting coats in that the selected assembly site is not demarcated by deposition of a GTP-bound small GTPase (*Ren et al., 2013*). The lipid phosphatidylinositol 4,5-bisphosphate (PtdIns(4,5)P$_2$) is required for

**eLife digest** Cells can take proteins and other molecules that are either embedded in, or attached to, their surface membrane and move them inside via a process called endocytosis. This process often involves a protein called clathrin working together with numerous other proteins. Early on, a complex of four proteins, called the adaptor protein-2 complex, interacts with both the 'cargo' molecules that are to be taken into the cell, and the cell membrane. Clathrin molecules then assemble into an ordered lattice-like coat, on top of the adaptor protein complex layer. This deforms a small patch of the cell membrane and curves it inwards. The clathrin molecules coat this pocket as it grows in size, until it engulfs the cargo. The pocket quickly pinches off from the membrane to form a bubble-like structure called a vesicle, which is brought into the cell.

A family of proteins termed Muniscins were thought to be involved in the early stages of endocytosis and have to arrive at the membrane before the adaptor protein-2 complex and clathrin. But experiments to test this idea—that reduced, or 'knocked-down', the production of Muniscins— had given conflicting results. As such, it remained unclear how the small patches of membrane carrying cargo molecules are marked as being destined to become clathrin-coated vesicles.

Now Umasankar et al. have studied the role that these proteins play in the early stages of endocytosis in human cells grown in a laboratory. A gene-editing approach was used to precisely disrupt a gene that codes for a Muniscin protein called FCHO2. Umasankar et al. observed that these 'edited' cells formed clathrin coats that were more irregular compared with those that form in normal cells. Nevertheless, clathrin-mediated vesicles still formed when this protein was absent, though the process of endocytosis was slower.

Similar results were seen when Umasankar et al. used the same approach to disrupt the gene for a related protein called FCHO1 in the same cells. A short fragment of the Muniscin proteins, called the linker, was shown to bind to, and activate, the adaptor protein-2 complex. The linker then recruits this complex to the specific regions of the cell membrane where clathrin-coated vesicles will form. Several dozen other proteins also accumulate where clathrin pockets form; as such, one of the next challenges will be to investigate if this mechanism of locally activating the cargo-gathering machinery is common in living cells.

clathrin assembly (*Boucrot et al., 2006*; *Zoncu et al., 2007*; *Nunez et al., 2011*), but this phosphoi-nositide is broadly distributed over the inner leaflet of the plasma membrane. Ordered polymerization of a clathrin patch in fact appears to be a variable and probabilistic process (*Ehrlich et al., 2004*; *Taylor et al., 2011*; *Cocucci et al., 2012*; *Brach et al., 2014*). The nucleation stage displays the widest temporal variation and likely involves the largest number of protein cofactors (*Taylor et al., 2011*). While the principal heterotetrameric cargo-selective adaptor, designated AP-2, is certainly important (*Mitsunari et al., 2005*; *Cocucci et al., 2012*; *Aguet et al., 2013*), its initial occurrence at inchoate bud sites is paralleled by a sizable suite of early-arriving pioneers, including eps15, intersectin, epsin, Fcho1 and Fcho2, CALM (*Taylor et al., 2011*) and Necap 1 (*Ritter et al., 2013*). The Fcho family (<u>Fer/Cip4 homology only</u> proteins), collectively termed the muniscins (*Reider et al., 2009*), is posited to act as internalization site founders on the basis of the biochemical properties of the proteins and the endocytic consequences of RNAi-mediated transcript silencing (*Henne et al., 2010*). Here we show that munsicins are indeed consequential endocytic pioneers, but that clathrin-mediated endocytosis still persists in their absence (*Uezu et al., 2011*; *Mulkearns and Cooper, 2012*; *Mayers et al., 2013*; *Brach et al., 2014*). We delineate a key biochemical aspect of the early operation of these endocytic proteins that regulates the rate and extent of clathrin-lattice fabrication at the cell surface.

## Results

FCHO1 (KIAA0290; located on chromosome 19) and FCHO2 (on chromosome 5) display ~50% overall amino acid identity and a similar domain organization (*Figure 1A*; *Katoh, 2004*). Both are EFC domain proteins (*Shimada et al., 2007*) characterized by a C-terminal μ-homology domain (μHD), distantly related in primary sequence but with an analogous fold to the cargo-engaging μ2 subunit of the hetero-tetrameric AP-2 clathrin adaptor (*Reider et al., 2009*). This combination of an EFC domain (alternatively designated the F-BAR domain [*Frost et al., 2008*; *Heath and Insall, 2008*]) with a μHD is unique, and

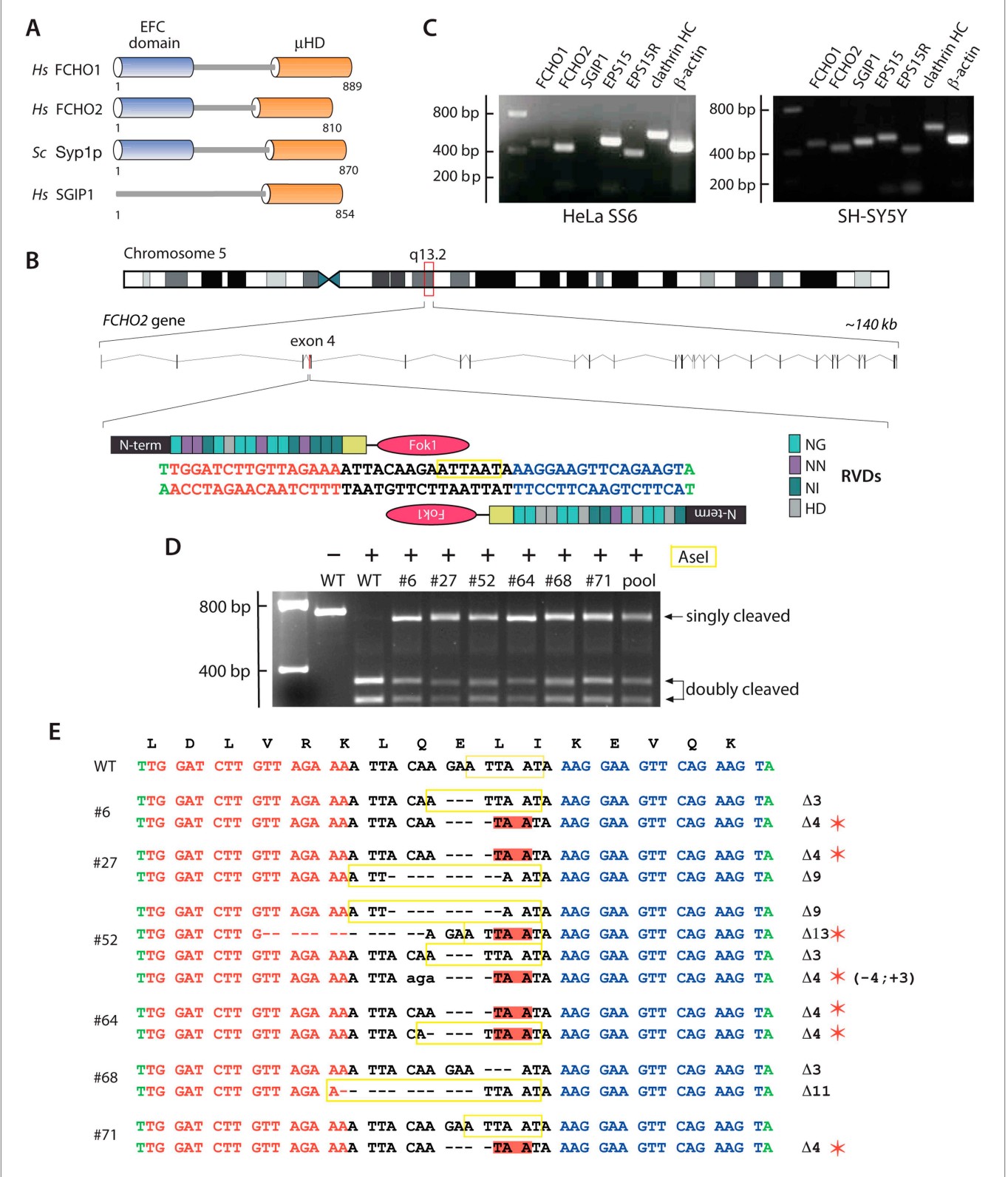

**Figure 1**. Gene editing the *FCHO2* locus in HeLa cells. (**A**) Domain arrangement of *Homo sapiens* (*Hs*) and *S. cerevisiae* (*Sc*) muniscin family proteins. The crescent-shaped EFC domain (Extended FCH (Fer/Cip4 homology) domain) is alternatively designated the F-BAR domain due to an overall structural homology of the α-helical anti-parallel EFC dimer to the BAR (Bin1/amphiphisin/Rvs) domain family of proteins. (**B**) Chromosomal location

*Figure 1. Continued on next page*

*Figure 1. Continued*

and genomic organization of the *FCHO2* gene with pertinent details of TALEN design. The repeat variable di-residues (RVD) selective for the different deoxyribonucleotides are color-coded (single letter amino acid notation). The endogenous AseI recognition sequence within the targeted exon is boxed (yellow). (**C**) Gene-specific RT-PCR analysis of various endocytic protein and control mRNA transcripts in the parental HeLa SS6 and neuroblastoma SH-SY5Y cells. HC; heavy chain. (**D**) AseI restriction enzyme digestion of *FCHO2* gene-specific PCR amplicons from genomic DNA extracted from wild-type (WT) and TALEN-treated clones. The undigested parental (HeLa) PCR product and digested PCRs are shown. The 'pool' designates a PCR reaction from a genomic DNA sample of TALEN-transfetced HeLa cells prior to clone selection. The AseI nuclease generates three PCR DNA fragments; the 55-bp band is not visible on these gels but causes the shift in the singly-cleaved product to 645 bp. (**E**) Genomic sequence analysis of TALEN clones. TALEN generated insertions (lower case letters) and deletions are indicated in relation to the WT nucleotide and amino acid sequences. AseI restriction sites are boxed (yellow) and in-frame stop codons are highlighted (red) and identified with a red asterisk.

these paralogous proteins have orthologues in chordates, arthropods and nematodes (*Katoh, 2004*; *Umasankar et al., 2012*; *Mayers et al., 2013*), as well as in unicellular eukaryotes (*Boettner et al., 2009*; *Reider et al., 2009*; *Stimpson et al., 2009*). In the budding yeast *Saccharomyces cerevisiae*, Syp1p has an analogous architecture to FCHO1/2 despite limited (<20%) overall primary sequence identity (*Figure 1A*). Phylogenetic dendrograms (TreeFam TF328986) (*Ruan et al., 2008*) show that a third chordate member, Sgip1, is more closely related to FCHO2 than to FCHO1, but this neuronal protein, while containing a modular C-terminal μHD, lacks the folded N-terminal helical EFC domain (*Figure 1A*; *Uezu et al., 2007*; *Dergai et al., 2010*). At steady state, endogenous FCHO2 (*Henne et al., 2010*; *Umasankar et al., 2012*) and both transfected FCHO1 (*Reider et al., 2009*; *Taylor et al., 2011*; *Umasankar et al., 2012*) and Sgip1 (*Uezu et al., 2007*; *Stimpson et al., 2009*) colocalize extensively with clathrin-coated structures, unlike several other EFC domain proteins (Cip4/Fbp17/Pacsin) implicated in endocytosis (*Modregger et al., 2000*; *Kamioka et al., 2004*; *Taylor et al., 2011*).

We used transcription activator-like effector nuclease (TALEN)-mediated gene editing to address a lack of coherence and important functional discrepancies in the literature (*Henne et al., 2010*; *Nunez et al., 2011*; *Uezu et al., 2011*; *Cocucci et al., 2012*; *Mulkearns and Cooper, 2012*; *Umasankar et al., 2012*) that could be due to the extent of, or variability in, Fcho1/2 transcript silencing by short-lived synthetic siRNAs. The *FCHO2* gene was targeted first (*Figure 1B*) since it is widely expressed (*Katoh, 2004*; *Lundberg et al., 2010*; *Uhlen et al., 2010*; *Uezu et al., 2011*; *Borner et al., 2012*; *Mulkearns and Cooper, 2012*) and FCHO2 is readily detected on immunoblots of HeLa lysate (*Henne et al., 2010*; *Uezu et al., 2011*; *Umasankar et al., 2012*). RT-PCR with gene-specific primers identifies appropriate amplicons for *FCHO2*, *EPS15*, *EPS15R*, the clathrin heavy chain, and the control β-actin genes in HeLa SS6 cells (*Figure 1C*). Available RNA sequencing (RNA-seq) data (*Lundberg et al., 2010*; *Uhlen et al., 2010*) corroborate *FCHO2* expression in HeLa cells. A tract within exon 4 of the *FCHO2* locus was selected for TALEN pair construction (*Figure 1B*). This targeted genomic region flanked by the assembled TALENs contains an endogenous AseI restriction site and the mRNA encodes residues Leu93–Ile98 of the α3a helix in the folded EFC domain (*Henne et al., 2007*).

After selection, an AseI resistant ~650-bp PCR fragment, in addition to the wild-type 351-, and 294-bp cleavage products, is evident in six representative HeLa TALEN clones (*Figure 1D*). The digests of the individual clones are similar to the PCR products seen in the initial TALEN-transfected population pool. Although this pattern suggests only heterozygosity, sequencing of the PCR amplified alleles discloses several homozygous gene-disrupted HeLa lines (*Figure 1E*); some of the small deletions, although producing frame-shifted nonsense mutations, regenerate an AseI restriction site (*Figure 1E*). One of the expanded clones (#52) contains four distinct disrupted alleles, indicating a mixed cell population. Immunoblotting verifies the genotype of the clones (*Figure 2A*).

Following RNAi, the phenotype typical of FCHO2-depleted HeLa cells is a reduced surface clathrin density and apparently enlarged or clustered clathrin-coated structures (*Figure 2B,C*) (*Mulkearns and Cooper, 2012*; *Umasankar et al., 2012*; but see *Cocucci et al., 2012*; *Henne et al., 2010*). Confocal optical sections of the *FCHO2*-disrupted clones illustrate that a very similar phenotype is evident in gene-edited cells. Compared with the control HeLa cells (*Figure 2D*), four of the edited lines (clones #6, 27, 52 and 64) have a clearly abnormal arrangement of AP-2 and EPS15-positive clathrin-coated structures (*Figure 2E–I*). The exceptions (*Figure 2J,K*) are clone #68 and #71, which contains one undisrupted allele (*Figure 1E*), illustrating that there is no apparent gene dosage effect (*Henne et al., 2010*).

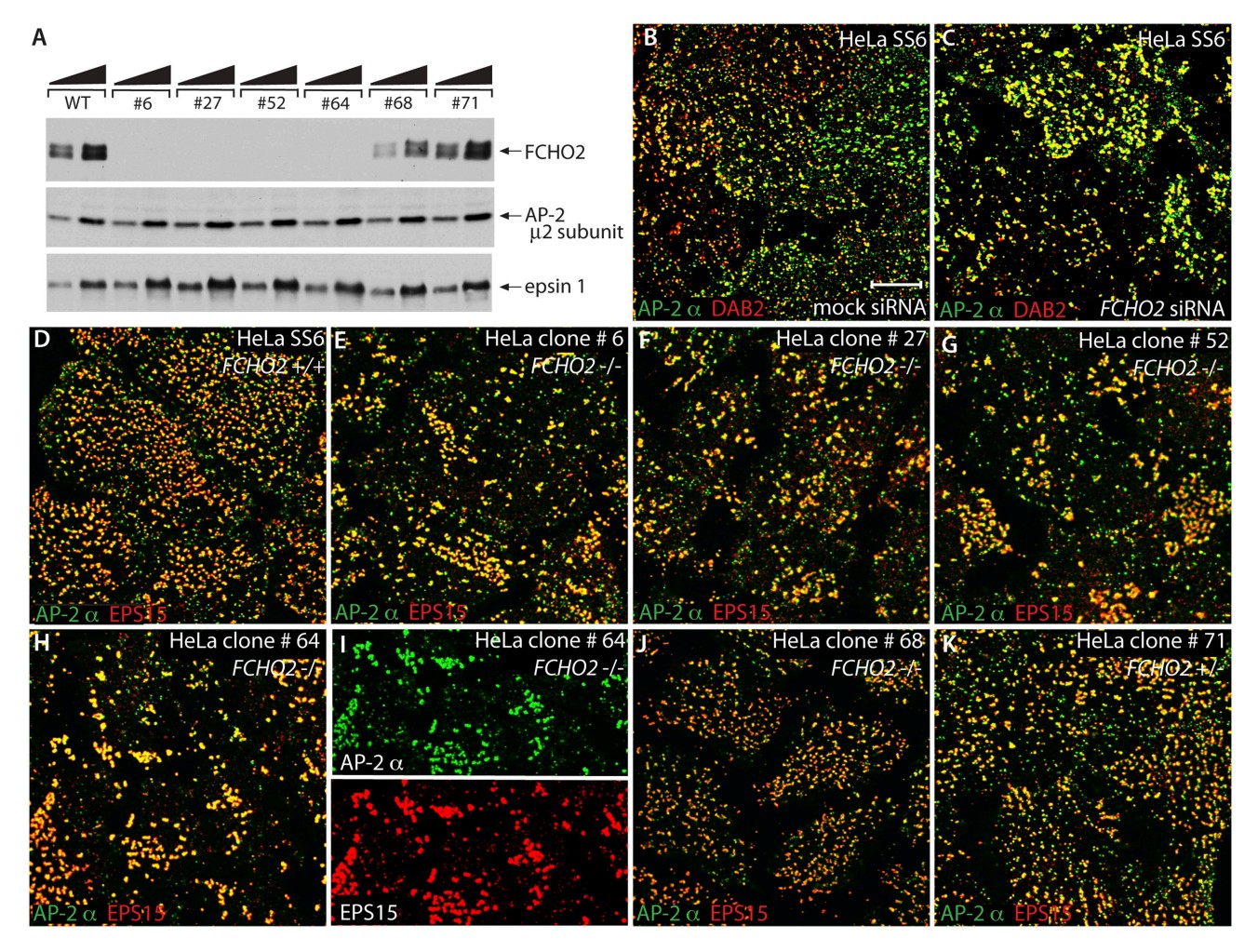

**Figure 2**. *FCHO2*-nullizygous HeLa cells exhibit abnormal clathrin-coated structures at the plasma membrane. (**A**) Whole cell lysates from wild-type (WT) HeLa SS6 cells and TALEN clones #6, #27, #52, #64, #68 and #71 were analyzed by SDS-PAGE. Duplicate immunoblots were probed with polyclonal antibodies directed against FCHO2, the AP-2 μ2 subunit and epsin 1. (**B** and **C**) Selected but representative confocal optical sections of HeLa SS6 cells either mock transfected (**B**) or transfected with *FCHO2* transcript-targeting siRNA oligonucleotides (*Umasankar et al., 2012*) (**C**). Fixed cells were stained with a mAb directed against the AP-2 α subunit (AP.6, green) and affinity purified antibodies against DAB2 (red). (**D–K**) HeLa SS6 cells (**D**) or the indicated TALEN-treated clones (**E–K**) were fixed and stained with mAb AP.6 (green) and affinity purified antibodies directed EPS15 (red). Color-separated channels from a portion of the micrograph of clone #64 cells (**H**) are presented (**I**). Scale bar: 10 μm.

We selected clone #64 for further detailed analysis as this cell has two distinct 4-bp deletions, each leading to an immediate in-frame stop codon (Q97X) terminating the FCHO1 polypeptide within the EFC domain (*Figure 1E*). RT-PCR shows a strong qualitative decrease in FCHO2 mRNA abundance in clone #64 cells (*Figure 3A*), presumably due to nonsense-mediated decay. In these FCHO2-null cells, the relative levels of numerous other endocytic and some unrelated cellular proteins are essentially unaltered (*Figure 3B*). Notably, FCHO1, undetectable by immunoblotting in HeLa cells (*Uezu et al., 2011*; *Umasankar et al., 2012*; *Figure 3B*) does not show a compensatory increase in clone #64 cells, either at the RNA or protein level, clearly dissimilar to the counteracting expression reported in BS-C-1 cells (*Nunez et al., 2011*). Endogenous FCHO1 is readily detectable, along with FCHO2, in K562 erythroleukemia cells. RNA-seq results indicate an eightfold higher *FCHO1* expression in K562 cells compared with HeLa (*Lundberg et al., 2010*; *Uhlen et al., 2010*). Moreover, RT-PCR fails to detect evidence of the SGIP1 transcript in either HeLa or clone #64 cells (*Figure 3A*). Indeed, SGIP1 is essentially a neuronally-expressed protein (*Trevaskis et al., 2005*; *Uezu et al., 2007*), with RT-PCR

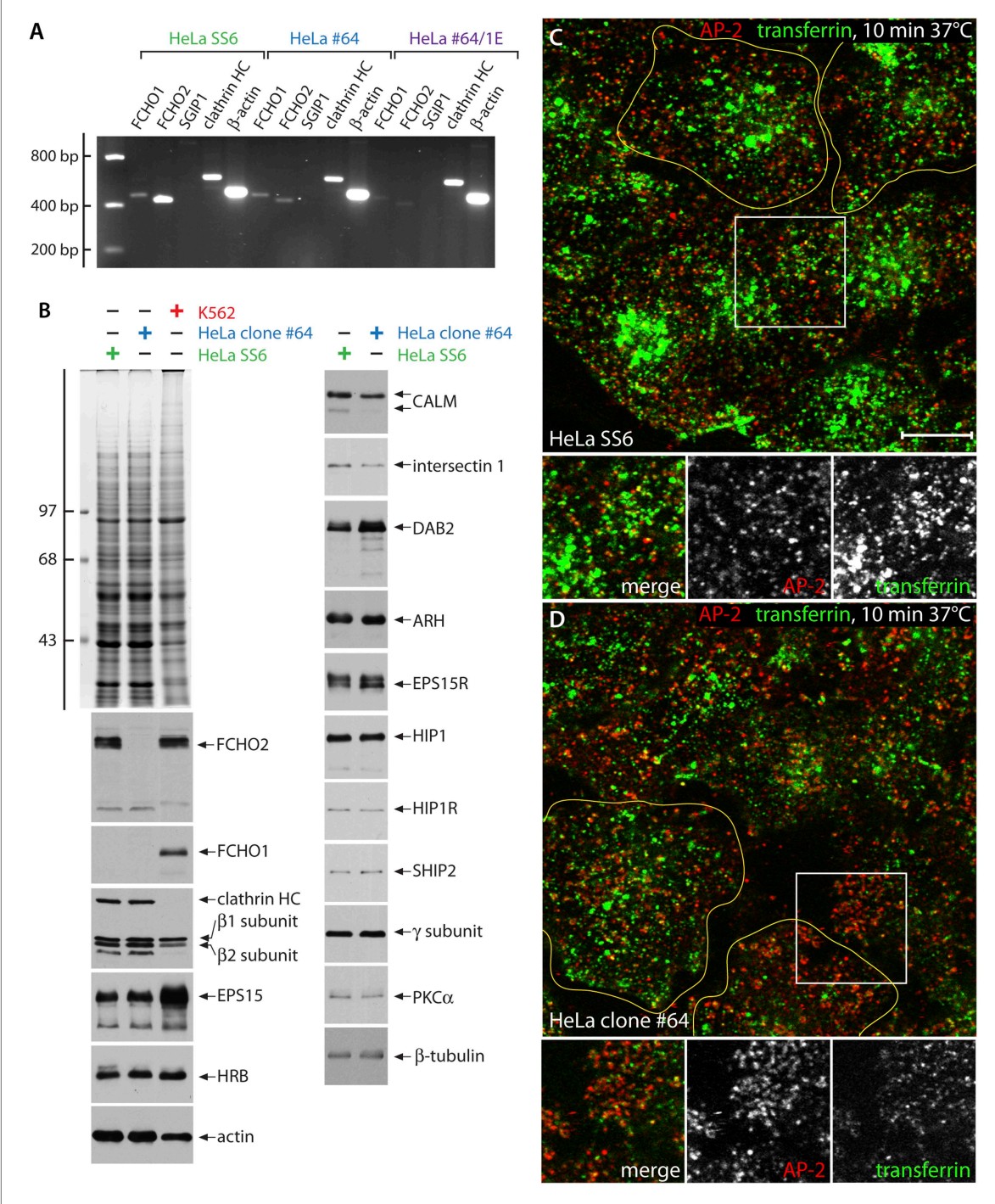

**Figure 3**. Characterization of FCHO2-null HeLa clone #64 cells. (**A**) Semi-quantitiative RT-PCT analysis of muniscin protein transcripts in parental HeLa SS6, clone #64 and clone#64/1.E cells. The same PCR primers as in *Figure 1C* were used. HC; heavy chain. (**B**) Whole cell lysates from HeLa SS6, clone #64 and K562 cells were resolved by SDS-PAGE and either stained with Coomassie blue or transferred to nitrocellulose. Replicate blots were probed with antibodies specific for the indicated proteins. Positions of the molecular mass standards (in kDa) are indicated on the left. (**C** and **D**). After incubation at 37°C for 60 min in serum-free medium, cover slip-attached HeLa SS6 (**C**) or clone #64 cells (**D**) were pulsed for 10 min with Alexa Fluor488-labelled transferrin (green). After chilling on ice and washing with ice-cold PBS, the cells were fixed and stained for AP-2 (red) with the α-subunit specific mAb AP.6. Representative single confocal optical sections focused on a medial region, rich in transferrin-positive endosomal structures, are shown, with color-separated channels of the boxed regions shown below. Scale bar: 10 μm.

(*Figure 1C*) and RNA-seq (*Lundberg et al., 2010*; *Uhlen et al., 2010*) indicating >30-fold higher transcript abundance in SH-SY5Y neuroblastoma cells vs HeLa cells.

The spatial reconfiguration of clathrin-coated structures in clone #64 cells is accompanied with altered kinetics of transferrin receptor internalization. 10 min after addition of fluorescent transferrin, bright juxtanuclear clusters of recycling endosomes are present in the parental HeLa SS6 cells (*Figure 3C*). At this juncture, little of the transferrin colocalizes with AP-2. By contrast, after 10 min of identical treatment, there is still residual overlap between the added transferrin and surface-associated AP-2 in the clone #64 cells, while the intensity of transferrin within endosomes is obviously diminished and occurs in a more peripheral early endosome compartment (*Figure 3D*). These results concur that internalization is obviously rate limiting in these edited cells (*Uezu et al., 2011*; *Mulkearns and Cooper, 2012*) and a diffuse pool of surface receptors not clustered within clathrin-coated structures is evident after a brief pulse of labeled transferrin. Cessation of receptor-mediated endocytosis (*Henne et al., 2010*) is not, however, apparent.

While clone #64 cells closely resemble *FCHO2 + FCHO1* gene-silenced HeLa cells (*Mulkearns and Cooper, 2012*; *Umasankar et al., 2012*), the presence of the appropriate *FCHO1* PCR product (*Figures 1C and 3A*) is consistent with full-length cDNA clones (CR597235, CR620485, and CR625849) isolated from HeLa cells, and also with the identification of FCHO1- and FCHO2-derived peptides in proteomic analysis of HeLa cell extracts (*Beausoleil et al., 2006*). Although transcript silencing of *FCHO1* alone has little effect on surface clathrin coat organization in HeLa SS6 cells (*Figure 4—figure supplement 1*), it is possible that the residual clathrin-coated structures present in clone #64 could be due to the expression of *FCHO1*.

## Stable clathrin assemblies at the plasma membrane of FCHO1 + FCHO2 gene-edited cells

An iterative TALEN-based approach to disrupt the *FCHO1* gene in clone #64 cells was therefore used. A tract within exon 5, encoding residues Lys24–Ser29 on the predicted EFC domain α2 helix (*Henne et al., 2007*) and housing an internal ApaI site, was selected for disruption (*Figure 4A*). Similar to the *FCHO2* TALEN lesions, the selected clones display an ApaI-resistant band(s) in addition to the two 387- and 150-bp cleavage products of the 537-bp PCR amplicon (*Figure 4B*). Also, parallel AseI digests confirm retention of the disrupted *FCHO2* alleles (*Figure 4C*). Sequencing of a representative clone (#64/45F1), uncovers three differently disrupted and a single wild-type *FCHO1* allele. Yet this does not represent a mixed cell population, as three independent subclones derived from this line all display the identical genotype (*Figure 4D*). Because HeLa cells are aneuploid (*Bottomley et al., 1969*), and chromosome 19 is duplicated in some HeLa lines (*Adey et al., 2013*; *Landry et al., 2013*), we interpret this to be TALEN-mediated disruption of three of the four *FCHO1* alleles in HeLa SS6 cells. We used clone #64/1.E (designated clone 1.E cells) for all further analysis. More generally, our results highlight potential complications when using gene editing on cultured cell lines with non-diploid genomes.

Genotype–phenotype analysis corroborates diminished steady-state mRNA levels for both *FCHO1* and *FCHO2* (*Figure 3A*). Biochemical pull-down assays further reveal exceptionally low levels of FCHO1 protein in clone 1.E cells compared with K562, which normally express this protein. Despite typical enrichment of cytosolic munscins by affinity isolation on immobilized GST-EPS15 (595–889) (*Umasankar et al., 2012*), there is no immunoreactivity for either protein in the TALEN-edited cells (*Figure 5A*). Still, the AP-2 in each of the cell extracts binds avidly to the GST-EPS15, while EPS15R remains within the supernatant fractions. These results thus resemble (neuronal) dynamin 1 expression in SK-MEL-2 cells; despite mRNA recognition there is no immunodetectable protein and this GTPase is ~70-fold less abundant than the ubiquitous dynamin 2 (*Grassart et al., 2014*). Yet, if ectopically expressed, either GFP-tagged full-length FCHO1 or Sgip1 are readily detectable in whole HeLa clone 1.E cell lysates (*Figure 5B*). By contrast, AP-2 and numerous other endocytic and trafficking proteins are present in equivalent amounts in HeLa and clone 1.E cells (*Figure 5B*).

Not surprisingly, the overall disposition of AP-2-positive endocytic structures in the clone 1.E cells (*Figure 5D*) differs from HeLa SS6 cells (*Figure 5C*) and resembles closely the clone #64 cells, from which they were derived (*Figure 2H*). And the clathrin-coated regions in the clone 1.E cells are populated with a full cohort of early-arriving clathrin coat pioneers despite the abnormal surface arrangement. At steady state, surface deposited AP-2 and clathrin are accompanied by CALM, DAB2, intersectin 1, EPS15 and HRB (*Figure 5C–J*), all members of the pioneer module of endocytic proteins (*Taylor et al., 2011*) and found in HeLa cell clathrin-coated vesicles (*Borner et al., 2012*). Thus, in

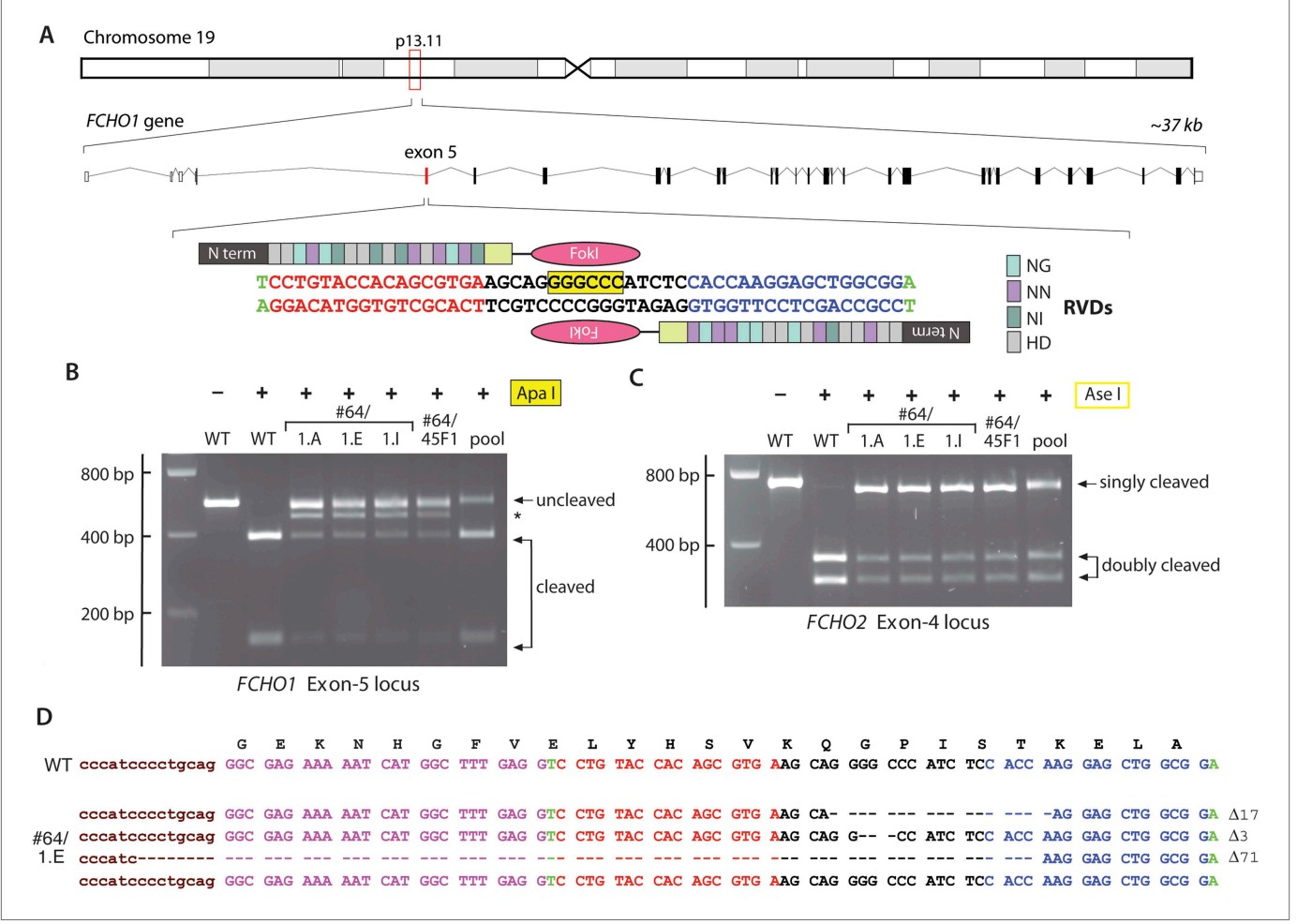

**Figure 4**. TALEN targeting of the *FCHO1* locus in HeLa clone #64 cells. (**A**) Chromosomal location and genomic organization of the *FCHO1* gene with pertinent details of TALEN design. The repeat variable diresidues (RVD) selective for the different deoxynucleotides are color-coded (single letter amino acid notation). The position of the internal ApaI restriction site within the targeted exon is highlighted (yellow). (**B**) ApaI restriction digest analysis of exon-specific *FCHO1* PCR products from parental wild-type (WT) HeLa SS6 and selected *FCHO1*-targeting TALEN pair transfected cell lines. The PCR product marked with an asterisk represents the 71-bp deletion allele in the #64/45F1 line and derived subclones. The pool PCR was prepared from genomic DNA isolated from the TALEN-transfected population of clone #64 HeLa cells before clone selection. (**C**) AseI digests of FCHO2 amplicons to verify the *FCHO2⁻/⁻* genotype of all the second- and third-round clones. (**D**) Sequencing results from PCR product amplified from genomic DNA extracted from clone #64/1.E cells. As indicated by the ApaI digest (**B**), all the clones derived from clone #64/45F1 have the identical genotype.

The following figure supplement is available for figure 4:

**Figure supplement 1**. FCHO1 transcript silencing does not alter the arrangement of surface clathrin-coated structures.

obvious contrast to other observations (*Henne et al., 2010*), neither clathrin nor AP-2 is completely soluble in the clone 1.E cells in spite of the extremely low level of muniscins compared with unedited cells. Further, these results demonstrate plainly that congregation of EPS15/EPS15R and intersectin 1 at bud zones does not depend upon stoichiometric FCHO1/2 (*Henne et al., 2010*). Our findings are congruent with results with yeast (*Boettner et al., 2009*; *Reider et al., 2009*; *Stimpson et al., 2009*) and nematode (*Mayers et al., 2013*) model systems that show deletion of the single muniscin expressed in these organisms is not lethal.

Comparative quantitative analysis of the distribution of AP-2 in HeLa SS6 and the clone 1.E cells indicates a sharp increase in the very smallest (<60 nm²) coated structures on the ventral surface. In the parental cells, these account for <25% of the AP-2 signal while in the FCHO1/2-depleted cells these small puncta represent almost 45% of the total AP-2. There is also an increase in the largest (>600 nm²)

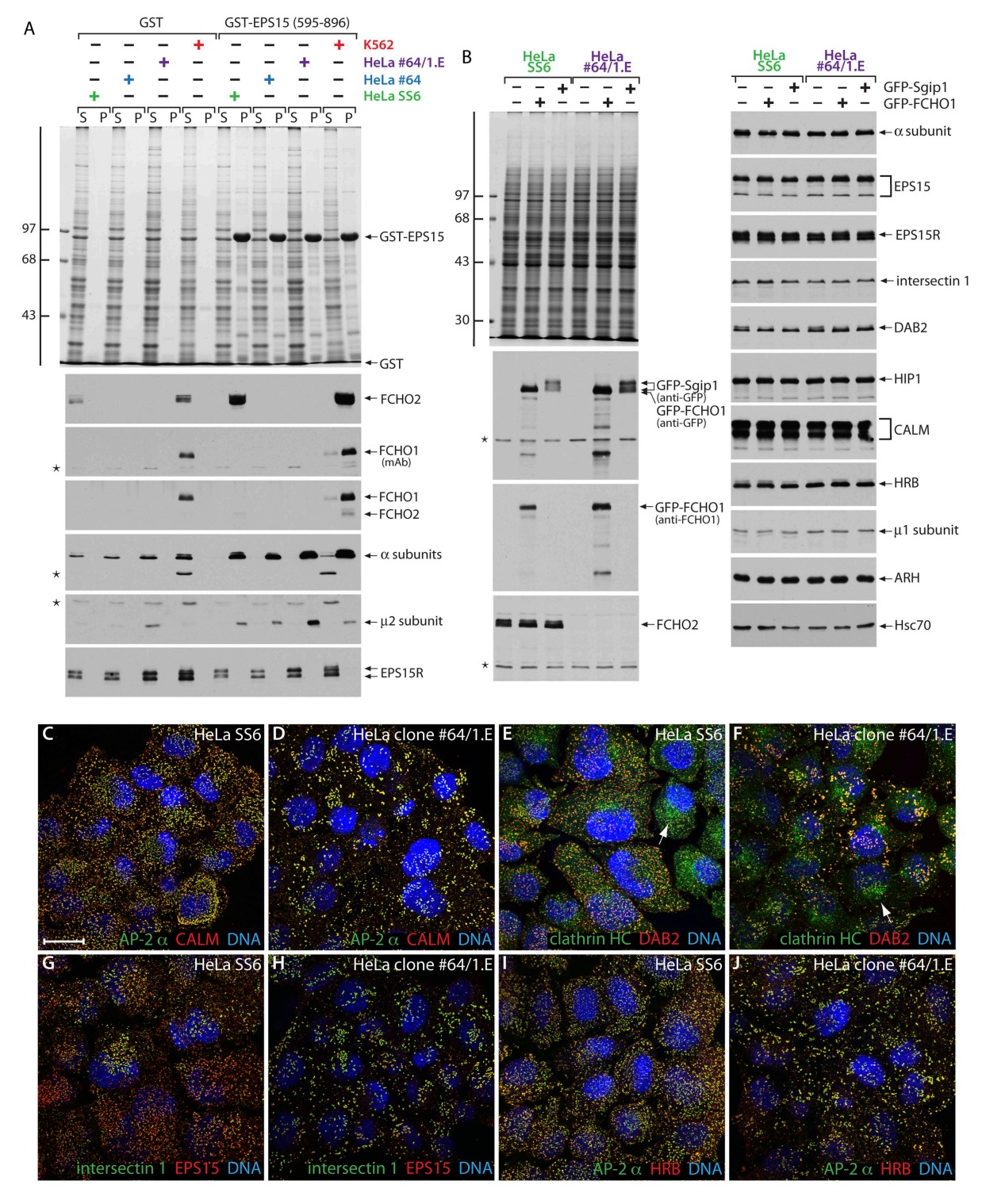

**Figure 5**. HeLa clone #64/1.E cells have undetectable levels of muniscin proteins. (**A**) Samples of 100 μg of GST or GST-EPS15 (595–896) prebound to glutathione-Sepharose beads were incubated with cell lysates from HeLa SS6, clone #64, clone #64/1.E and K562 cells. After washing, aliquots of the supernatant (S) and pellet (P) fractions were resolved by SDS-PAGE and stained with Coomassie blue or replicates transferred to nitrocellulose. Positions of the molecular mass standards (in kDa) are shown. Results from both an anti-FCHO1 mAb directed against the μHD and an affinity-purified polyclonal raised against the EFC domain and weakly cross-reactive with FCHO2 are shown. (**B**) Parental HeLa SS6 or clone #64/1.E cells, mock transfected or transfected with plasmid DNA encoding GFP-FCHO1 or GFP-Sgip1 as indicated, were collected after 16 hr and whole cell lysates subject to SDS-PAGE

*Figure 5. Continued on next page*

*Figure 5. Continued*

analysis. Gels were either stained with Coommassie blue or transferred to nitrocellulose. Duplicate blots were probed with designated antibodies. Positions of the molecular mass standards (in kDa) are shown. (**C–J**) Representative images of HeLa SS6 (**C**, **E**, **G**, **I**) or HeLa clone #64/1.E (**D**, **F**, **H**, **J**) cells after immunolabeling with anti-AP-2 mAb AP.6, affinity-purified anti-CALM antibodies, anti-clathrin heavy chain (HC) mAb X22, affinity purified anti-DAB2 antibodies, anti-intersectin 1 mAb , affinity-purified anti-EPS15 antibodies, and anti-HRB serum. Arrows indicate the Golgi/endosomal population of clathrin present in both control and clone #64/1.E HeLa cells. Scale bar: 10 μm.

clathrin assemblages, as previously seen in *FCHO2* silenced HeLa cells (***Mulkearns and Cooper, 2012***; ***Umasankar et al., 2012***). After disruption of the *FCHO1* and *FCHO2* genes, 46% of membrane-associated AP-2 is present in structures >600 nm$^2$ while only 29% of AP-2 occurs within this size category in the HeLa SS6 cells. Analogous results are obtained upon measuring the distribution of EPS15 in the two cell types. Engineered loss of muniscins therefore shifts the equilibrium of clathrin lattices at the cell surface to the very small and the largest forms.

The ultrastructural basis for the altered surface distribution of clathrin assemblies in clone 1.E cells can be appreciated in deep-etch EM replicas (***Figure 6***). At steady state, the ventral surface of the parental HeLa SS6 cell line typically exhibits a range of clathrin-coated assemblies and invaginated buds (***Figure 6A,B***). Both solitary coated buds and spherical coated invaginations adjacent to regions of planar clathrin lattice are regularly dispersed over the membrane. By contrast, conspicuous expanses of plasma membrane devoid of polyhedral clathrin typify the clone 1.E cells, and overall the number of deeply invaginated buds is reduced (***Figure 6C–E***). Whereas abutting planar assemblages are abundant, individual, spatially isolated de novo forming coated buds are present in clone 1.E cells, as well as small patches of flat polygonal assemblies (***Figure 6C,D***). Beside the spatial effects of *FCHO1/2* gene disruption, structural alterations within the assembled clathrin layer are also apparent. Higher magnification views reveal that, in contrast to the well ordered, principally hexagonal flat sheets in control HeLa cells (***Figure 6B***, ***Figure 6—figure supplement 1***), the clone 1.E cells display geometrically poorly ordered lattices (***Figure 6E***, ***Figure 6—figure supplement 1***). So, despite having a complement of co-assembled AP-2, EPS15, DAB2, CALM, intersectin 1 and HRB, the lattices do not appear to assemble as regularly as in control HeLa cells.

## Clathrin-mediated cargo internalization in clone 1.E cells

A two-minute pulse of fluorescent transferrin at 37°C labels surface AP-2 puncta and peripheral early endosomes in control HeLa cells (***Figure 7A***), as the unsynchronized ligand fluxes through the cell surface into the endosomal compartment. The clone 1.E cells have the larger, less regular clathrin-coated structures also labeled with transferrin at 2 min, although a diffuse pool of unclustered receptors is clearly evident on the cell surface (***Figure 7B***). This confirms endocytic entry is rate limiting in FCHO1/2-depleted cells (***Uezu et al., 2011***; ***Mulkearns and Cooper, 2012***). A buildup of transferrin receptors at the cell surface may be explained, in part, by lattice defects and a reduced number of invaginated buds. After 10 min, the bulk of transferrin is within endosomes in both the control (***Figure 7C***) and clone 1.E cells, but residual transferrin at the enlarged clathrin-coated structures and on the plasma membrane in the edited cells is apparent. (***Figure 7D***).

Given the coat protein densities within the aberrant clathrin-coated structures typical of clone 1.E cells, and the accumulation of transferrin receptors at the cell surface, the extent of transferrin clustering is reduced relative to the HeLa cell control line. Because the transferrin receptor has a YXXØ-type YTRF internalization signal that depends on AP-2 for uptake (***Hinrichsen et al., 2003***; ***Motley et al., 2003***; ***Huang et al., 2004***), our results suggest that AP-2 cargo capture may be compromised in the clathrin assemblies in clone 1.E cells. The disparity in transferrin sequestration between steady-state clone 1.E and ordinary clathrin-coated structures is better appreciated by transfecting GFP-tagged full-length FCHO1 into the gene-edited cells (***Figure 7E–L***). Forced reexpression of FCHO1 normalizes the AP-2 arrangement on the ventral surface (***Figure 7F***). A 2-min pulse of transferrin concentrates the ligand near these AP-2/FCHO1-positive spots (***Figure 7E–H***). Adjacent untransfected clone 1.E cells have the characteristic dispersed transferrin signal over the plasma membrane, and even strongly AP-2-positive structures do not concentrate transferrin similar to relatively dimmer AP-2 puncta in the transfected cells (***Figure 7E***). Further analysis of transferrin binding to GFP-FCHO1-transfected HeLa clone 1.E cells by total internal reflection microscopy (***Figure 7—figure supplement 1***) corroborates the defect in transferrin clustering in the clone 1.E cell line.

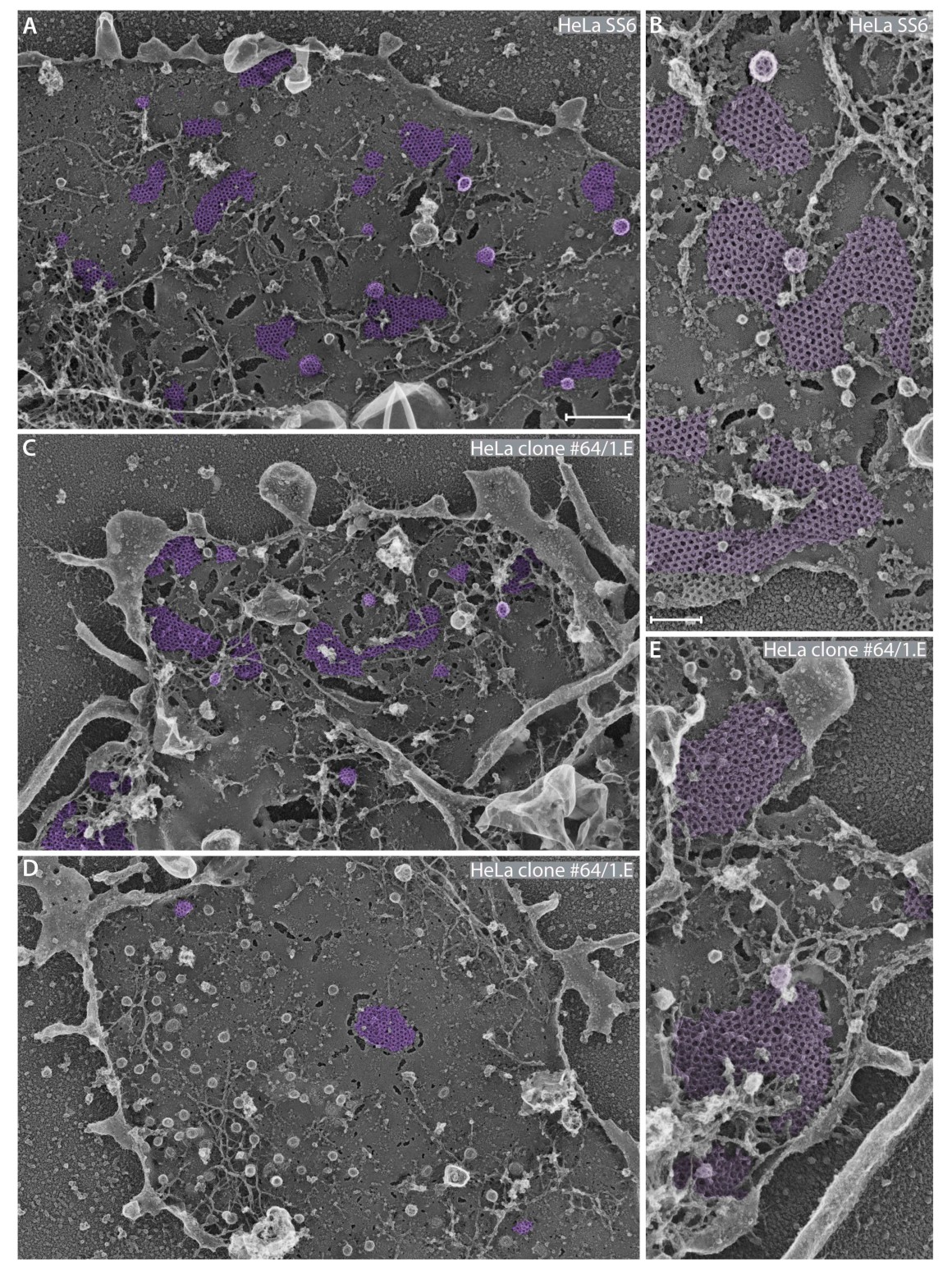

**Figure 6**. Ultrastructural analysis of gene-edited clone #64/1.E cell clathrin lattices. (**A–E**) Selected but representative deep etch-EM replicas revealing the glass-attached ventral surface of control HeLa SS6 (**A** and **B**) or clone #64/1.E (**C–E**) cells. Polyhedral clathrin assemblies are pseudocolored purple. Higher magnification views of control (**B**) and 1.E (**E**) cells highlight irregular assembly of clathrin trimers in the absence of FCHO1 and FCHO2. The

*Figure 6. Continued on next page*

*Figure 6. Continued*

numerous surface-attached spherical buds in (**D**) are caveolae not clathrin-coated structures; they completely lack a characteristic polyhedral coat. Scale bars: 500 nm in **A**, **C** and **D**; 100 nm in **B** and **E**.

The following figure supplement is available for figure 6:

**Figure supplement 1**. Lattice assembly defects in the FCHO1/2-depleted HeLa clone #64/1.E cell line.

After a 10-min pulse of transferrin, medial plane confocal optical sections illustrate that both the clone 1.E cells and neighboring GFP-FCHO1-transfected cells accumulate the ligand in perinuclear endosomes (*Figure 7I–L*), albeit internalization is more efficient in the presence of expressed FCHO1. Stalled ligand-bound receptor still persists at the surface of the clone 1.E cells. Overall, the data obtained from the TALEN-edited cells are fully consistent with the siRNA results in HeLa cells (*Mulkearns and Cooper, 2012*; *Uezu et al., 2012*; *Umasankar et al., 2012*). We conclude that clathrin-coat assembly and endocytosis does not cease entirely when cellular levels of muniscin proteins are extremely low. These proteins also do not appear obligatory for sustained clathrin lattice growth (*Cocucci et al., 2012*), but rather impact the fidelity of the polyhedral construction.

## Muniscin-dependent restoration of a regular clathrin coat distribution in clone 1.E cells

Transiently expressed GFP-FCHO1 clusters at surface puncta and changes the general AP-2 configuration compared with the surrounding non-transfected clone 1.E cells (*Figure 7F,G, 8B,B′* and *Figure 8—figure supplement 1*). By contrast, transfected GFP alone is diffusely cytosolic, and has no effect on the abnormal clathrin patches in the gene-edited cells (*Figure 8A,A′*). This effect of FCHO1 on clathrin coat arrangement is not restricted to gene-edited lines. The native topology of surface clathrin lattices in cultured MCF-7 cells, a breast adenocarcinoma, is even more exaggerated and extensively abutting than the gene-edited clone 1.E cells (*Figure 8—figure supplement 2*). A considerable size distribution of clathrin assemblies can thus occur naturally in different cell types (*Maupin and Pollard, 1983*; *Gaidarov et al., 1999*; *Akisaka et al., 2003*; *Ehrlich et al., 2004*; *Grove et al., 2014*; *Sochacki et al., 2014*). Strikingly, temporary introduction of GFP-FCHO1 (or GFP-FCHO2) into MCF-7 cells switches the AP-2 assemblies to considerably smaller puncta (*Figure 8—figure supplement 2*). Because the altered clathrin distribution in the clone 1.E (and MCF-7) cells is extremely penetrant and robust, we used complementation transfection experiments to delineate the domain(s) of FCHO1 required to revert the morphology to the wild-type pattern.

On expressing alone the lipid-binding EFC domain (*Henne et al., 2007*; *Edeling et al., 2009*) in clone 1.E cells, while membrane associated, it neither clusters at clathrin-coated structures nor modifies their appearance (*Figure 8C,C′*). By contrast, the GFP-µHD does concentrate at plasma membrane clathrin assemblages, because the domain functions as an interaction hub (*Reider et al., 2009*; *Dergai et al., 2010*; *Henne et al., 2010*; *Mulkearns and Cooper, 2012*; *Umasankar et al., 2012*) but, again, does not refashion these atypical coated zones (*Figure 8D,D′*). Yet expression of proteins containing the polypeptide stretch linking the EFC domain to the µHD reestablishes a more regular arrangement of AP-2-positive sites in clone 1.E cells (*Figure 8E–G′*, *Figure 8—figure supplement 1*). Systematic truncation and deletion experiments better pinpoints the active region of the FCHO1 linker (*Figure 8I*). In various proteins, inclusion of FCHO1 residues 339–416 promotes the redistribution of AP-2 into more a uniform and evenly dispersed form over the ventral plasma membrane. Deletion of residues 316–467 from full-length GFP-tagged FCHO1 prevents complementation of the clathrin distribution despite deposition of the transfected protein at clathrin pincta (*Figure 8H,H′*). The linker section of FCHO1, but neither the EFC domain nor µHD, also repatterns the bulky clathrin in MCF-7 cells to more regularly dispersed arrays (*Figure 8—figure supplement 2*). Forced expression of the N-terminal linker-containing segment of Sgip1 (residues 1–514) has a similar consequence in clone 1.E cells, in spite of the absence of an EFC domain (*Figure 8J,J′*).

Together, these experiments show that the central linker in FCHO1, FCHO2 and Sgip1, despite likely being intrinsically disordered, is required to revert the organization of clathrin-coated structures to the more uniform format typical of control cells. A remarkable aspect of this activity is that at very low expression levels, the linker can alter the steady-state organization of coated regions without being enriched at assembly zones, and can even operate when expressed as a soluble protein.

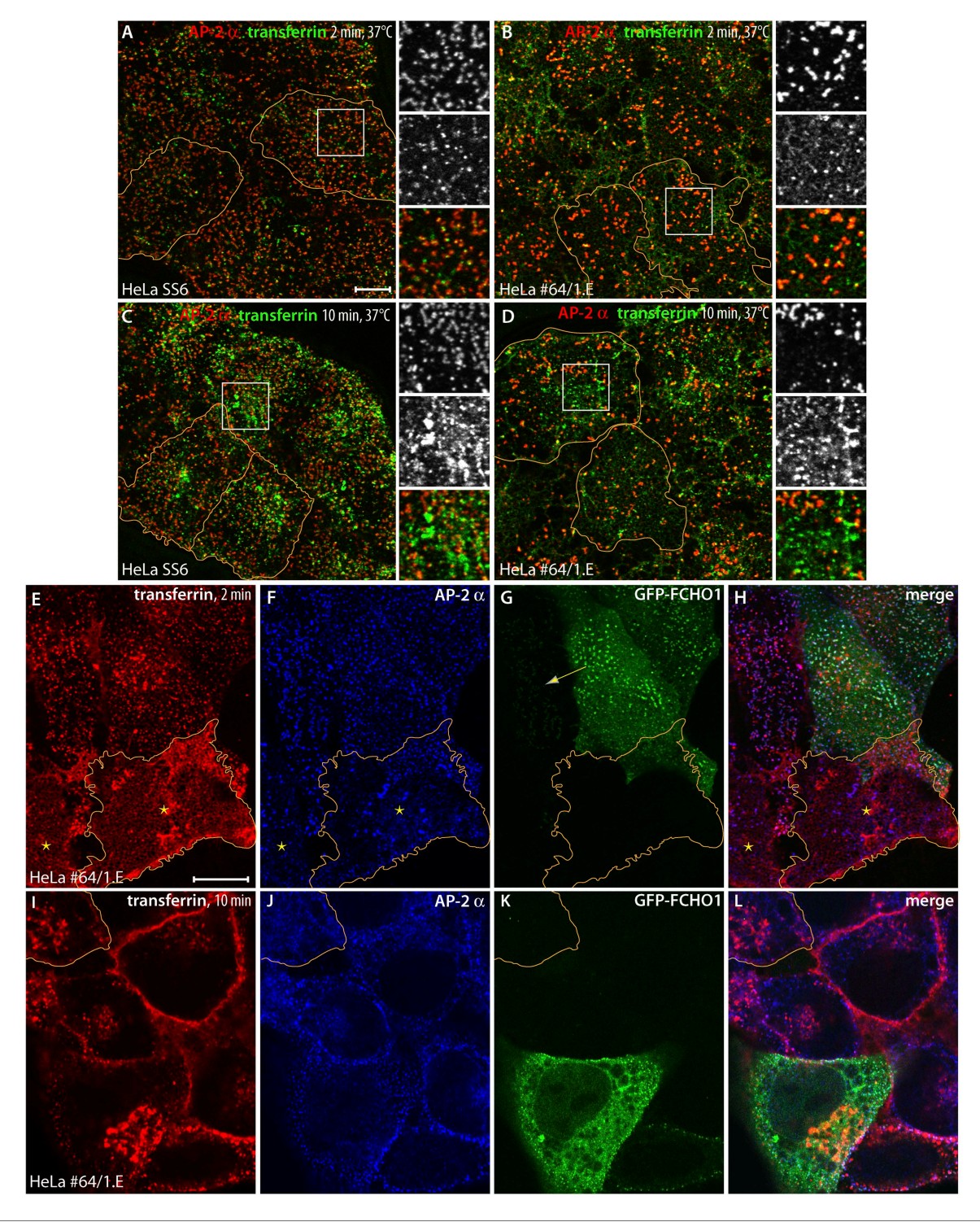

**Figure 7**. Clathrin-dependent cargo internalization in HeLa clone #64/1.E cells. (**A** and **B**) Representative confocal optical sections of HeLa SS6 (**A**) or clone #64/1.E (**B**) cells incubated with 25 µg/ml Alexa Fluor488-conjugated transferrin (green) for 2 min at 37°C before washing on ice. Fixed cells were stained with the anti-AP-2 α subunit mAb AP.6 (red). The borders of some cells in the field are outlined (orange) and color-separated views of the boxed regions are shown on the right. Scale bar (for **A–D**): 10 µm. (**C** and **D**) Analogous images from control HeLa SS6 (**C**) or clone #64/1.E cells (**D**) after a 10-min pulse of transferrin. (**E–H**) Confocal image of adherent ventral region of HeLa clone #64/1.E cells transiently transfected with GFP-FCHO1 (1-889) (green) before addition of a 2-min pulse of 25 µg/ml Alexa Fluor568-labeled transferrin (red) as in A and **B**. Fixed cells were immunolabeled for

*Figure 7. Continued on next page*

*Figure 7. Continued*

AP-2 with mAb AP.6 (blue). A cell in each field is outlined (orange) and a low GFP-FCHO1 expressing cell (arrow) and adjacent untransfected FCHO1/2-depleted cells (asterisks) are indicated. Scale bar (for **E**–**L**): 10 μm. (**I**–**L**) Medial plane optical section of GFP-FCHO1 (1–889) transfected HeLa clone #64/1.E cells following a 10-min pulse of transferrin. The perimeter of a non-complemented clone #64/1.E cell is indicated (orange).

The following figure supplement is available for figure 7:

**Figure supplement 1**. Reconstitution of transferrin capture within clathrin-coated structures in HeLa clone #64/1.E cells.

## Muniscins as potential allosteric activators of AP-2

The linker tract is the most divergent portion of the muniscin family members, with generally <25% identity overall. Yet alignments of this region of Fcho1, Fcho2, Sgip1 (*Katoh, 2004*) reveal they share small blocks of phylogenetically conserved residues, particularly rich in acidic side chains and phos-pho-modifiable residues (*Figure 9A*). As the linker regions from FCHO1, FCHO2 and Sgip1 can each correct the HeLa clone 1.E cell coat morphology, it seems likely that these blocks of common residues are functionally important. In fact, post-translational phosphorylation events are enriched at protein–protein interaction surfaces (*Nishi et al., 2011*), and high-throughput mass spectrometry identifies numerous phosphosites within the shared region of the linker domain (*Figure 9A*).

Several distinct lines of evidence indicate that the central linker acts with AP-2: First, the anomalous transferrin clustering at coated patches noted in clone 1.E cells indicates muniscins could impact AP-2 cargo engagement. Second, in GST pull-down assays, the minimal FCHO1 linker (residues 316–467) binds physically to cytosolic AP-2 (*Figure 9B*). All four subunits of the heterotetramer sediment with the GST-FCHO1 linker, but engagement of AP-2 is not as efficient as the C-terminal portion of the cargo-selective monomeric adaptor ARH that binds directly to the β2 subunit appendage (*He et al., 2002*; *Mishra et al., 2005*), and clathrin binding is limited. We find no evidence for the linker polypeptide engaging lipid modifying enzymes (PIP5KIγ [*Figure 9B*], PLD, DGKδ1 [not shown]) that could, in principal, affect clathrin coat assembly without physically being enriched at coat assembly zones. Fused to GST, the conserved central segments of FCHO2 (residues 314–444) and Sgip1 (residues 77–214) also bind to cytosolic AP-2, although the interaction with FCHO2 is comparatively weaker (*Figure 9—figure supplement 1A*).

The independently folded α- and β2-subunit appendages of AP-2 (*Figure 9C*) can engage FCHO1 (*Umasankar et al., 2012*). Yet titrating an excess of α appendage into pull-down assays does not alter the extent of soluble AP-2 heterotetramer engagement, so the major contact surface for the AP-2 on the FCHO1 linker tract does not appear to interact with the α appendage (*Figure 9—figure supplement 1B*). In the same experiment, the added α appendage does bind to the C-terminal region of EPS15 and concomitantly diminish cytosolic AP-2 association, in a dose-dependent manner (*Figure 9—figure supplement 1B*).

Binary interaction assays reveal that the purified tetrameric core of AP-2 binds physically to the linker regions of FCHO1, FCHO2 and Sgip1 (*Figure 9D*), with relative affinities similar to interactions with intact cytosolic AP-2 (*Figure 9—figure supplement 1A*). The sequence-related linkers of the muniscin members thus contact the AP-2 core directly. The other currently known interaction partner for the FCHO1/2 linker segment is Necap 1 (*Ritter et al., 2013*). Soluble Necap 1 in brain cytosol binds only very weakly to the GST-FCHO1 linker and not to the analogous portions of FCHO2 and Sgip1 (*Figure 9—figure supplement 1A*). However, when this association is assayed using immobilized GST-Necap 1 PHear domain (residues 1–133) and K562 cell lysates, a strong interaction with FCHO1 is apparent (*Figure 9—figure supplement 1C*). This suggests either that Necap 1 in brain cytosol is in an inhibited conformation or that the optimal binding site upon the FCHO1 linker lies outside of the minimal region (residues 316–467) necessary for correction of the clone 1.E cell clathrin phenotype.

A clue to the mechanistic connection between FCHO1/2 and AP-2 comes from additional biochemical studies. The muniscin μHD is a densely wired endocytic interaction hub, physically contacting the pioneer proteins Eps15/R, intersectin, Dab2, Hrb and CALM (*Reider et al., 2009*; *Henne et al., 2010*; *Mulkearns and Cooper, 2012*; *Umasankar et al., 2012*). The C-terminal portion of EPS15 is the highest affinity μHD binding partner (*Figure 5A*). A limited sequence tract between residues 595 and 636 represents the principal interaction region in EPS15 (*Umasankar et al., 2012*). Deletion of part of

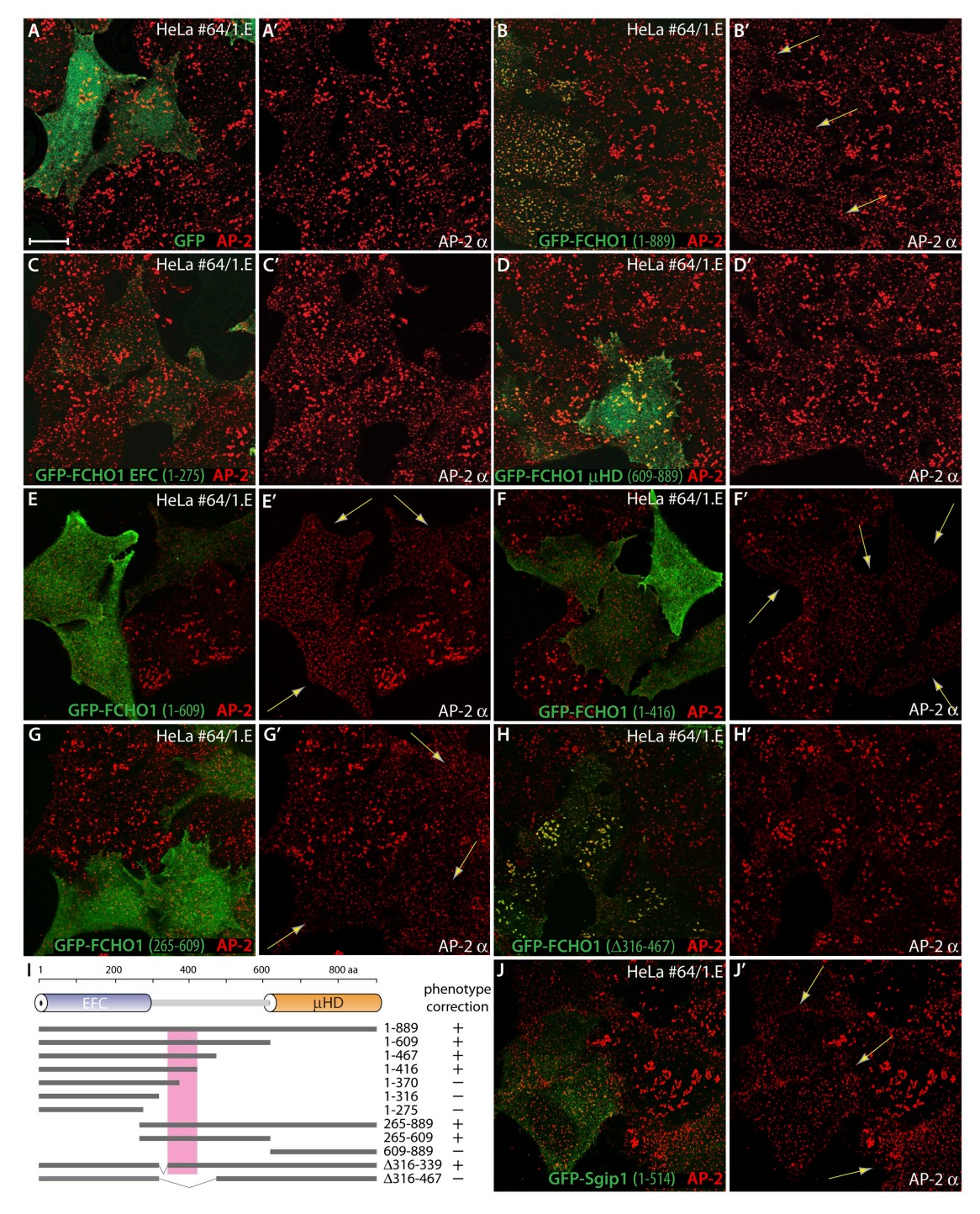

**Figure 8**. The muniscin unstructured linker sector regulates clathrin-coated structure topology. (**A–H′**) Representative single confocal optical sections of HeLa clone #64/1.E cells transiently transfected with GFP (**A** and **A′**), GFP-FCHO1 (1–889) (**B** and **B′**), GFP-FCHO1 EFC domain (1–275) (**C** and **C′**), GFP-FCHO1 µHD (609–889) (**D** and **D′**), GFP-FCHO1 (1–609) (**E** and **E′**), GFP-FCHO1 (1–416) (**F** and **F′**) GFP-FCHO1 (265–609) (**G** and **G′**) or GFP-FCHO1 (1–889; Δ316–467) (**H** and **H′**). Fixed cells were stained for AP-2 using the anti-AP-2 α-subunit mAb AP.6. Merged channel images (**A′–H**) and the corresponding AP-2 α subunit channel alone (**A′–H′**) are shown. Transgene-dependent refashioning of the irregular 1.E cell clathrin-coated structures to a more uniformly dispersed pattern is indicated (arrows). Scale bar for all panels: 10 µm. (**I**) Cartoon diagram of the overall domain organization of FCHO1 with the relative locations of the various N-terminally-tagged truncation and deletion constructs tested shown schematically. The minimal region

*Figure 8. Continued on next page*

*Figure 8. Continued*

necessary to correct the surface clathrin morphology is boxed. (**J–J'**) Selected confocal section of HeLa clone #64/1.E cells transfected with GFP-Sgip1 (1–514) analyzed as in (**A–H'**).
The following figure supplements are available for figure 8:
**Figure supplement 1**. Comparative expression of GFP-tagged FCHO1 protein fragments in HeLa cells.
**Figure supplement 2**. Oversized and clumped clathrin-coated structures in MCF-7 cells are normalized by ectopic FCHO1 or FCHO2 expression.

this region (residues 617–636) within the GST-EPS15 (595–896) fusion protein has a strong inhibitory effect on both FCHO1 and FCHO2 interactions from K562 cell lysates (*Figure 9E*). AP-2 also binds to the C-terminal third of EPS15 (*Figure 5*, *Figure 9—figure supplement 1B*), but with lower apparent affinity and utilizing the tandemly arrayed Asp-Pro-Phe (DPF) tripeptide repeats positioned distal to the µHD-binding region (*Benmerah et al., 1996*; *Iannolo et al., 1997*). Consequently, the association of cytosolic AP-2 with the GST-EPS15 (Δ617–636) is not different from the binding to the whole GST-EPS15 (*Figure 9E*). Yet there is a strong decrease in the amount of clathrin associated with the immobilized Δ617–636 fusion, which parallels the diminished engagement of FCHO1 and FCHO2. The majority of the soluble clathrin remains in the Δ617–636 assay supernatant fractions. Because AP-2 cannot engage clathrin trimers productively without undergoing allosteric activation (*Kelly et al., 2014*), it is as if FCHO1/2 promote conformational rearrangement of AP-2 to facilitate clathrin binding.

## Functional linkage between the muniscin linker polypeptide and AP-2

The role of the FCHO1 linker polypeptide was further explored using a tailored fusion to the cytosolic aspect of Tac (CD25), a type I transmembrane protein (*Uchiyama et al., 1981*). Transient expression of Tac in HeLa SS6 cells leads to surface accumulation because this α chain of the IL-2 receptor is poorly internalized (*Humphrey et al., 1993*). At steady state, individual low-level-expressing cells have a prominent surface pool of Tac and an internal population that overlaps partly with the Golgi-region marker GPP130 (*Linstedt et al., 1997*) and with the *trans*-Golgi network (TGOLN2/TGN46). The intracellular population in these cells thus represents primarily the biosynthetic trafficking of new Tac molecules en route to the surface (*Figure 10A,A'*). Cells expressing high levels of the Tac transgene have, in addition, a second internal accumulation of Tac, which colocalizes well with a pulse of internalized transferrin (*Figure 10A*). Switching a Tac-linker (FCHO1 residues 265–609) hybrid-encoding plasmid for Tac leads to similar but not identical expression and spread in HeLa SS6 cells (*Figure 10B–B″*). Increased localization of Tac-linker with the transferrin endocytic marker indicates that this customized fusion is more rapidly internalized from the cell surface than the native Tac protein. Analogous results are obtained upon expressing a Tac-FCHO1 µHD (residues 609–889) hybrid (*Figure 10C,C'*). Consequently, we find that ectopic Tac and Tac-fusion proteins display a dichotomous distribution of surface and internal pools, the composition of the latter differing depending on the level of Tac protein driven by the transfected plasmid DNA and on the amino acid sequence of the cytosolic portion.

Two conspicuous changes in clathrin coat organization occur when just the full FCHO1 linker is expressed in HeLa SS6 cells in a membrane-tethered form. First, assembled AP-2, normally restricted entirely to puncta on the inner leaflet of the plasma membrane, becomes mistargeted to juxtanuclear regions (*Figure 11A–A″*), located similarly to the Golgi and endosome compartment markers that overlap with intracellular Tac. The ectopic AP-2 is accompanied by colocalized EPS15 (*Figure 11A*). We infer that the high local density of the Tac-FCHO1 linker, as it traverses the biosynthetic pathway following ER export or concentrates on maturing endosomes, promotes anomalous recruitment of AP-2. Yet, when a corresponding Tac fusion with the FCHO1 µHD (residues 609–889) is transfected, very strong EPS15 and intersectin 1 (not shown) deposition on the Golgi/endosome region occurs, but strikingly this is not paralleled by AP-2 binding (*Figure 11B–B″*). The effect of overexpression of a Tac-linker + µHD (FCHO1 residues 316–889) hybrid appears similar to the linker alone, albeit with more robust EPS15 binding accompanying juxtanuclear AP-2 (*Figure 11C–C″*). Visualizing the Tac-FCHO1 fusion-expressing cells with an anti-Tac mAb (*Figure 11D–D″*) clarifies that the juxtanuclear pool of endocytic proteins is indeed recruited to Tac-positive internal structures. Thus, EPS15 (and intersectin 1) recruited by the Tac-anchored FCHO1 µHD concentrated on Golgi/endosome membranes does not

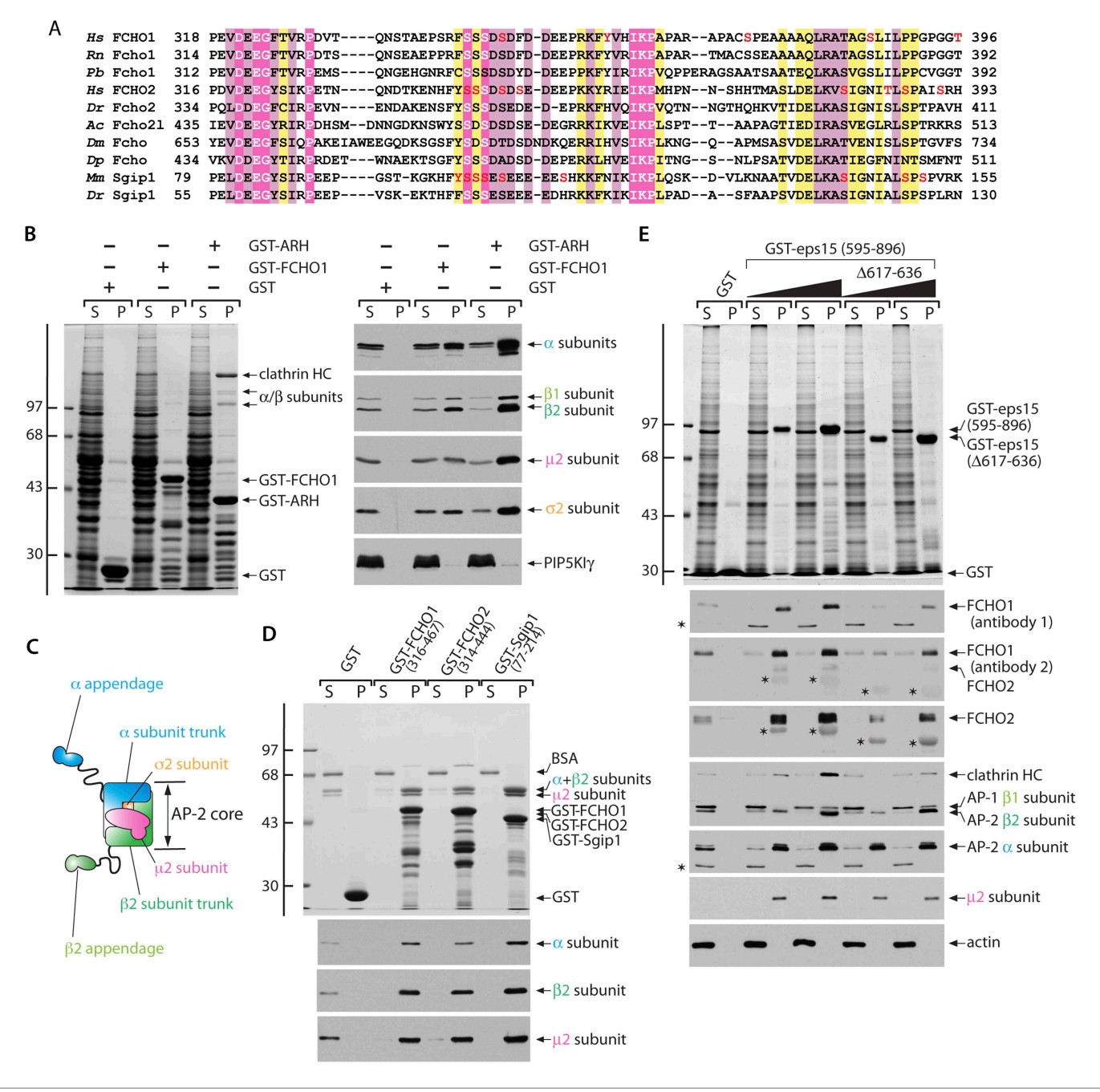

**Figure 9**. Functionally significant phylogenetic conservation within the muniscin central linker domain. (**A**) Muniscin–AP-2 interactions. T-Coffee (**Notredame et al., 2000**) generated multiple sequence alignment of the phylogenetically conserved linker region within muniscin members. Amino acid regions of Fcho1 from selected species: *Homo sapiens* (*Hs*; NP_001154829), *Rattus norvegicus* (*Rn*; XP_006252925), *Python bivittatus* (*Pb*; XP_007436694), Fcho2: *Hs* (NP_620137), *Danio rerio* (*Dr*; NP_001018617), Fcho2-like (Fcho2l): *Aplysia californica* (*Ac*; XP_005111676), the single FCHO member in *Drosophila melanogaster* (*Dm*; NP_001097723), and *Daphnia pulex* (*Dp*; EFX87825), as well as Sgip1: *Mus musculus* (*Mm*; NP_659155) and *Dr* (XP_005165952) are shown with appropriate residues numbers indicated. Identical residues are highlighted in magenta, highly similar residues in pale pink and conservatively substituted amino acids in yellow. The location of mass-spectrometry authenticated phosphosites (**Hornbeck et al., 2012**) are shown (red font). (**B**) Samples of 100 μg of GST, GST-FCHO1 (316–467) or GST-ARH (180–308) immobilized on glutathione-Sepharose were incubated with rat brain cytosol, washed and samples of each supernatant (S) and pellet (P) fraction separated by SDS-PAGE. Replicate gels were either stained with Coomassie blue (left) or immunoblotted with the designated antibodies (right). The positions of the molecular mass standards (in kDa) are indicated on the left. (**C**) Schematic illustration of the overall chain and domain composition of the AP-2 adaptor complex. (**D**) Samples of 100 μg of GST, GST-FCHO1

*Figure 9. Continued on next page*

*Figure 9. Continued*

(316–467), GST-FCHO2 (314–444) or GST-Sgip1 (77–214) immobilized on glutathione-Sepharose were used in pull-down assays with the purified AP-2 heterameric core complex as in (**B**). The identity of the large subunit trunk polypeptides (α and β2 subunits) and the myc-tagged μ2 subunit is confirmed on immunoblots with mAb clone 8, mAb 100/1 and mAb clone 31, respectively. (**E**) Aliquots of 100 μg of GST or either 25 μg or 100 μg of GST-EPS15 (595–896) or GST-EPS15 (595–896/Δ617–636) immobilized on glutathione-Sepharose were incubated with soluble lysate from K562 cells. After washing, portions of the supernatant and pellet fractions were analyzed as in (**B**). FCHO1 antibody 1 is a mAb while antibody 2 is an affinity-purified antibody that also recognizes FCHO2 weakly. Non-specific cross-reactive bands are indicated with asterisks; reactivity of the GST-EPS15 fusion proteins with the anti-FCHO1 and anti-FCHO2 antibody preparations is also indicated with asterisks. Notice the decreased FCHO1 and FCHO2 binding upon deletion of the minimal μHD sequence within the EPS15 C-terminal region that correlates with reduced clathrin association.
The following figure supplement is available for figure 9:

**Figure supplement 1**. Cytosolic AP-2 and the Necap 1 PHear domain (residues 1–133) bind to FCHO1 and FCHO2.

attract cytosolic AP-2, but AP-2 associated with these compartments through the membrane-linked linker polypeptide results in deposition of EPS15. Impressively, the massing of AP-2 and EPS15 upon perinuclear biosynthetic and endocytic organelles in Tac-FCHO1 linker (*Figure 12A–C*) or Tac-FCHO1 linker + μHD (*Figure 12D–F*) expressing cells occurs in the absence of any conspicuous PtdIns(4,5)$P_2$ enrichment in these compartments, as judged by a GFP-tagged PLCδ1 PH domain sensor selective for PtdIns(4,5)$P_2$.

Second, the Tac-linker fusion, when located on the plasma membrane, reconfigures the apparent dimensions of the clathrin-coated structures on the ventral surface relative to adjacent untransfected HeLa SS6 cells (*Figures 11C and 12A*). This overall shift to small, more uniformly grouped puncta also happens with transfection of the Tac-linker + μHD (*Figures 11D″ and 12D*), but not with the Tac-μHD fusion (*Figure 11B″*), even though the relocation of EPS15 is so dramatic in these cells that the protein becomes depleted from the surface AP-2-positive clathrin assemblies.

The phenotypic consequence of forced expression of the tailored Tac-FCHO1 chimeras is even more notable in the muniscin-depleted clone 1.E cells. While Tac alone has no noticeable effect on the characteristic distribution of AP-2 and EPS15 in these cells (*Figure 13A–D*), the Tac-FCHO1 linker promotes deposition of a fraction of these proteins onto juxtanuclear membranes and remodels the abnormal surface clathrin patches into more homogenous and regularly scattered assemblies (*Figure 13E–H*). By contrast, the Tac-FCHO1 μHD gives rise to noteworthy sequestration of EPS15 on juxtanuclear structures, as occurs in the parental HeLa SS6 cells, but again without accompanying AP-2 recruitment or refashioning of the surface clathrin structures in the clone 1.E cells (*Figure 13I–L*). Transient expression of the Tac-FCHO1 linker + μHD amalgamates the separate effects of the Tac-linker and Tac-μHD proteins, with marked AP-2 and EPS15 intracellular congregation and a drop in the dimensions of clathrin-coated puncta on the plasma membrane (*Figure 13M–P*). Parallel experiments utilizing the variously transfected HeLa clone 1.E cells and stained for Tac expression (*Figure 13—figure supplement 1*) recapitulate all these findings. That the ectopic transmembrane anchored FCHO1 linker alters the morphologic characteristics of surface clathrin patches toward uniform small puncta strengthens the proposition that enlarged coats in the clone 1.E cells originate directly from limiting cellular muniscin concentrations.

Crucially, the discrete clathrin/AP-2 surface assemblies that arise in Tac-linker-expressing cells are operational. Live-cell imaging of a HeLa SS6 cells stably transfected with a YFP-tagged AP-2 β2 subunit (β2-YFP) (*Keyel et al., 2008*) shows that forced expression of the Tac-FCHO1 linker (residues 265–609) in these cells again reconfigures the overall arrangement of clathrin-coated structures (*Figure 14A*). Using total internal reflection fluorescence microscopy, the more uniform and evenly spread β2-YFP puncta at the ventral surface of Tac-linker-containing cells differ from the adjacent untransfected cells (*Figure 14A*). After 2 min at 37°C, added fluorescent transferrin rapidly clusters at the β2-YFP-containing structures (*Figure 14B*) but, again, the relative ratio of transferrin to YFP differs between the Tac-linker producing and untransfected cells (*Figure 14A* vs *Figure 14B*), suggesting that the AP-2 spots in the Tac-FCHO1 linker-expressing cells are better able to concentrate transferrin receptors.

In the time-lapse data set, comparing the initial (2 min) transferrin signal (pseudocolored green) with a frame acquired 10 min later (colored blue) illustrates that the Tac-transfected cells, have more dynamic clathrin-coated vesicle uptake (*Figure 14C*). Numerous green or blue puncta are present in the transfected cells (*Figure 14C–E*), indicating that these transferrin-laden structures

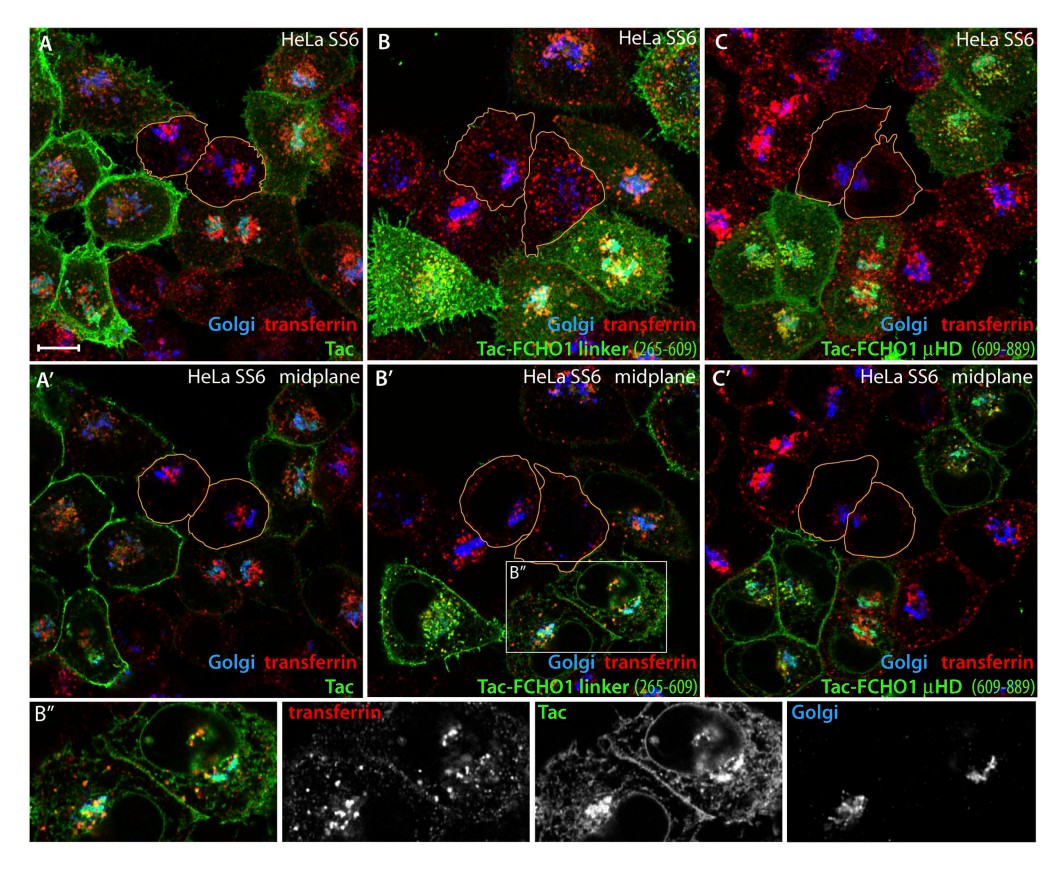

**Figure 10**. Trafficking of exogenous Tac and Tac-FCHO1 fusion proteins in HeLa cells. (**A–A'**) Maximal projection (**A**) and a selected medial plane (**A'**) of deconvolved confocal image z-stacks of HeLa SS6 cells transiently transfected with a Tac-encoding plasmid. Prior to fixation, the transfected cell population was pulsed with 25 µg/ml Alexa Fluor568 transferrin (red) for 10 min. Fixed cells were then stained with a mAb (7 G7B6) directed against the lumenal domain of Tac (green), anti-GPP130 antibodies (blue). Relative accumulation of the Tac protein in the Golgi (cyan color) and transferrin-positive endosomes (yellow) varies in different individual transfected cells. The cell perimeter of two selected, non-transfected cells is outlined (orange). Scale bar (for all panels): 10 µm. (**B–C**) Maximal projection (**B** and **C**) or selected medial planes (**B'** and **C'**) from deconvolved z-stacks acquired from HeLa SS6 cells transfected with either a Tac-FCHO1 linker (residues 265–610; **B**, **B'**) or a Tac-FCHO1 µHD (residues 609–889; **C**, **C'**). Fixed cells were processed identically to (**A**). Accumulation of he Tac-FCHO1 chimeras in the ER, as marked by nuclear envelope staining, is also apparent in some cells. The cell perimeter of two selected, non-transfected cells in each panel is outlined (orange). (**B"**) Enlargements of the color-separated channels from the boxed region in **B'** showing the colocalization of the transfected Tac-FCHO1 linker fusion with both the Golgi marker GPP130 and internalized transferrin.

disappeared/appeared between the two acquisition time points. A fluorescently conjugated anti-Tac antibody added after the time-lapse recording confirms that the cells exhibiting the smaller scattered β2-YFP structures are indeed expressing the Tac-linker chimera (*Figure 14F*). Also, at this time point transferrin concentrated within endosomes is evident in the evanescent field. We conclude that the linker polypeptide in munscin proteins affects the operation of AP-2 on the cell surface; the linker facilitates the production of more numerous cargo-filled clathrin-coated structures.

## Discussion

The polyhedrally interdigitated outer lattice is emblematic of the extensive protein–protein interactions that underpin the formation of a clathrin-coated vesicle. Yet perhaps the most impressive array of protein contacts occurs at the outset of clathrin polymerization on the limiting cell membrane. The early module of proteins implicated in the nucleation of surface clathrin coats is very highly interconnected

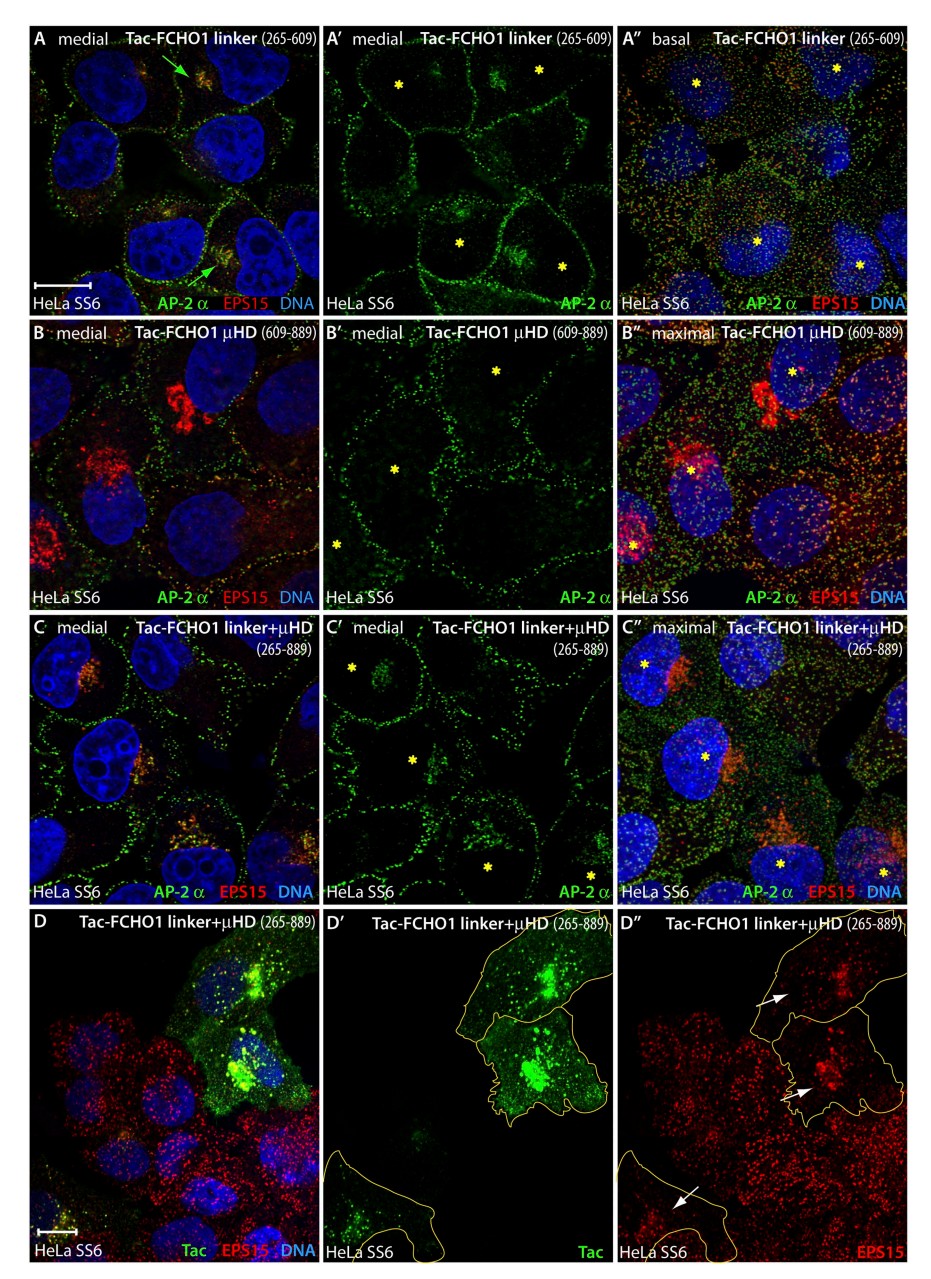

**Figure 11**. An artificial transmembrane FCHO1 linker protein misrecruits AP-2 onto internal membrane structures. (**A–A″**) Selected medial (**A** and **A′**) or basal (**A″**) optical sections of deconvolved confocal *z*-stacks collected from HeLa SS6 cells transiently transfected with Tac-FCHO1 linker (residues 265–609). Fixed cells were stained with anti-AP-2 α subunit mAb AP.6 (green) and affinity purified anti-EPS15 antibodies (red) and mounted in Hoechst 33342 to label DNA (blue) prior to imaging. Aberrant accumulation of AP-2 and EPS15 adjacent to the nucleus (arrows) in Tac-transfected cells (asterisks) correlates with smaller clathrin-coated puncta in the basal optical section of the same cell population (**A″**). Scale bar (for **A–C″**): 10 µm. (**B–C″**) Chosen middle (**B**, **B′**, **C** and **C′**) or maximal projection (**B″** and **C″**) optical sections of HeLa SS6 cells transfected with Tac-FCHO1 µHD (residues 609–889) or Tac-FCHO1 linker + µHD (residues 265–889) and prepared analogously to (**A–A″**). The Tac-fused µHD produces dramatic relocalization of EPS15 (**B**), diminishing this protein in surface AP-2-positive puncta (**B″**) yet AP-2 does not relocate similarly. With ectopic expression of the FCHO linker + µHD, prominent accumulation of irregular intracellular AP-2 and EPS15 in a juxtanuclear locations again correlates with diminished surface clathrin spots, which are again relatively deficient in EPS15 because of the massive deposition of this endocytic pioneer

*Figure 11. Continued on next page*

*Figure 11. Continued*

component upon intracellular membranes. (**D**–**D″**) Single ventral optical section of HeLa SS6 cells transfected with Tac-FCHO1 linker + µHD (residues 265–889) and stained with anti-Tac mAb (7 G7B6; green) and affinity-purified anti-EPS15 antibodies (red). Relocalization of EPS15 occurs only in Tac overexpressing cells (arrows), and the dose-dependent massing of EPS15 in the perinuclear region limits the amount of EPS15 present in surface clathrin-coated structures. The limiting membrane of the Tac-expressing cells is delineated (orange). Scale bar: 10 µm.

(*McMahon and Boucrot, 2011*; *Reider and Wendland, 2011*); microscopic temporal connections (*Taylor et al., 2011*) are enabled by nanoscopic macromolecular associations. The principal interaction hub is undoubtedly AP-2, which can, through the independently folded α and β2 appendages, bind to >20 other endocytic factors utilizing four separate contact interfaces (*Traub, 2009*; *Kelly and Owen, 2011*; *McMahon and Boucrot, 2011*). Numerous other pioneer proteins bind physically to PtdIns(4,5)$P_2$, similar to AP-2 (*Höning et al., 2005*; *Jackson et al., 2010*), and some contain EH and SH3 domains, producing a densely populated interactome to coordinate membrane remodeling, coat biogenesis and cargo selection. The inherent redundancy and functional overlap between several of these early-arriving proteins is responsible for the robust nature of clathrin-mediated endocytosis.

For example, CALM, a PtdIns(4,5)$P_2$ binding clathrin-associated sorting protein for the R-SNAREs VAMP2, -3 and -8 (*Koo et al., 2011*; *Miller et al., 2011*; *Sahlender et al., 2013*), interacts with the µHD of FCHO1 (*Henne et al., 2010*; *Umasankar et al., 2012*). CALM also contains an FXDXF-type interaction motif that engages directly the PHear domain of Necap 1 (*Ritter et al., 2007*), has several

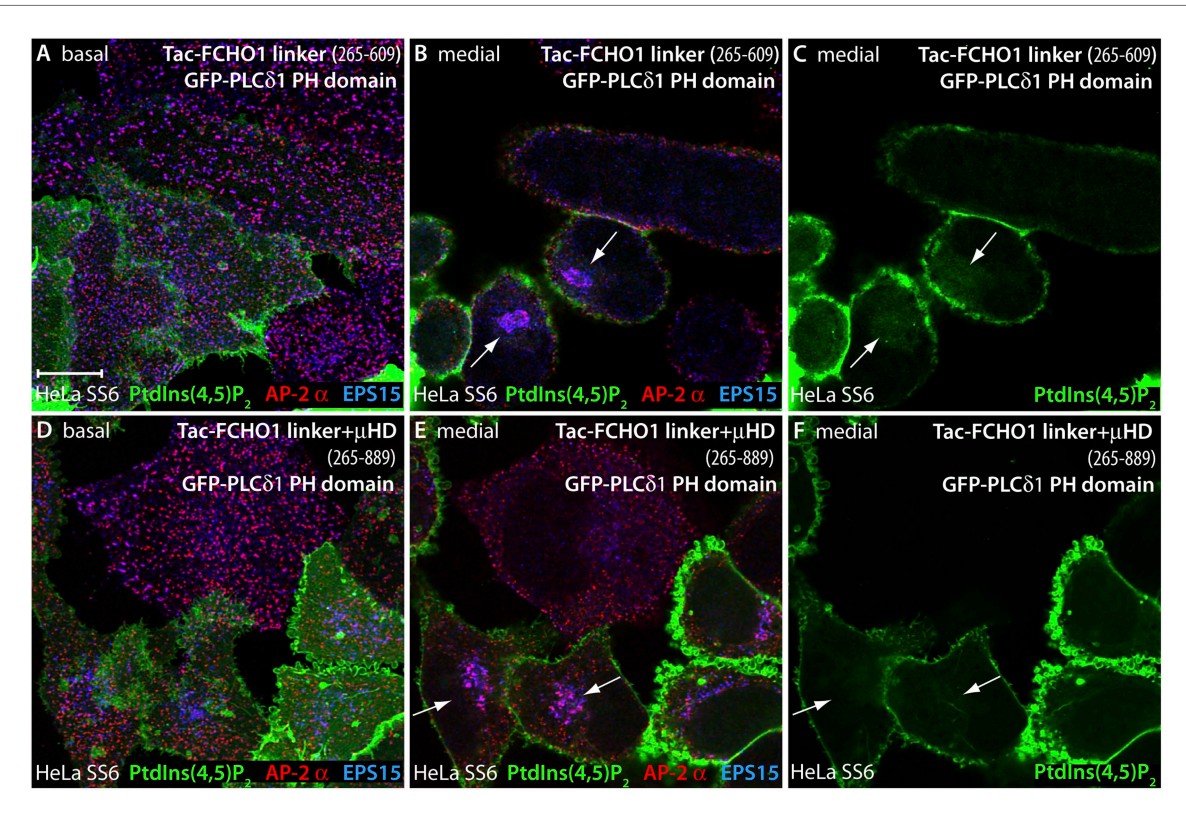

**Figure 12**. PtdIns(4,5)$P_2$ is not enriched at sites of intracellular AP-2 accumulation. (**A**–**F**) HeLa SS6 cells cotransfected with a mixture of either Tac-FCHO1 linker (residues 265–609) (**A**–**C**) or Tac-FCHO1 linker + µHD (residues 265–889) (**D**–**F**) and GFP-PLCδ1 PH domain encoding plasmids were fixed and stained with anti-AP-2 α subunit mAb AP.6 (red) and affinity purified anti-EPS15 antibodies (blue). The GFP fluorescence in the medial region of the Tac-linker expressing cells (**C**) represents the soluble pool of this fluorescent lipid probe adjacent to the nucleus. Note also the decrease in size and more regular arrangement of the AP-2- and EPS15-positive clathrin coated surface structures in the Tac-linker (**A**) and Tac-linker + µHD (**D**) expressing cells, identified by mislocalized deposition in a juxtanuclear position (arrows). Scale bar: 10 µm.

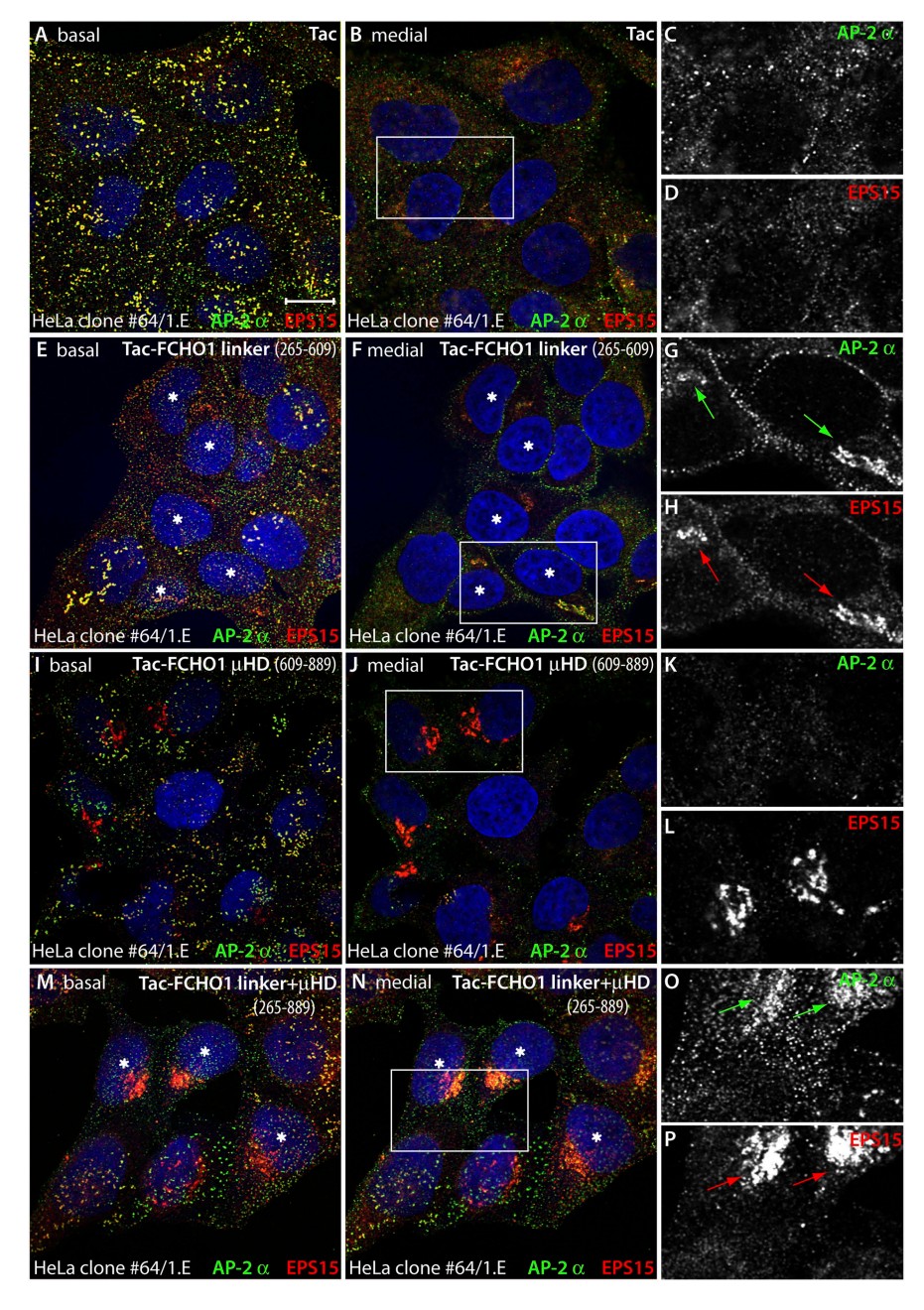

**Figure 13**. Forced expression of the Tac-FCHO1 linker fusion restores clathrin coat distribution in gene-edited HeLa cells. (**A–P**) Basal (**A**, **E**, **I**, **M**) and medial (**B**, **F**, **J**, **N**) confocal sections from deconvolved z-image stacks from HeLa clone #64/1.E cells immunolabeled with antibodies against AP-2 (mAb AP.6; green) and EPS15 (red) and Hoechst 33342 for DNA (blue). The clone #64/1.E cells were short-term transfected with Tac (**A–D**), Tac-FCHO1 linker (residues 265–609) (**E–H**), Tac-FCHO1 µHD (residues 609–889) (**I–L**) or Tac-FCHO1 linker + µHD (residues 265–889) (**M–P**) before fixation. Tac-expressing cells, identified by recruitment of AP-2/EPS15 onto internal membranes (asterisks) are indicated, and enlarged, color-separated views of the rectangular regions in the medial sections are presented on the right. Scale Bar: 10 µm.

The following figure supplement is available for figure 13:

**Figure supplement 1**. Mislocalized EPS15 in Tac-linker expressing HeLa clone #64/1.E cells overlaps with intracellular Tac.

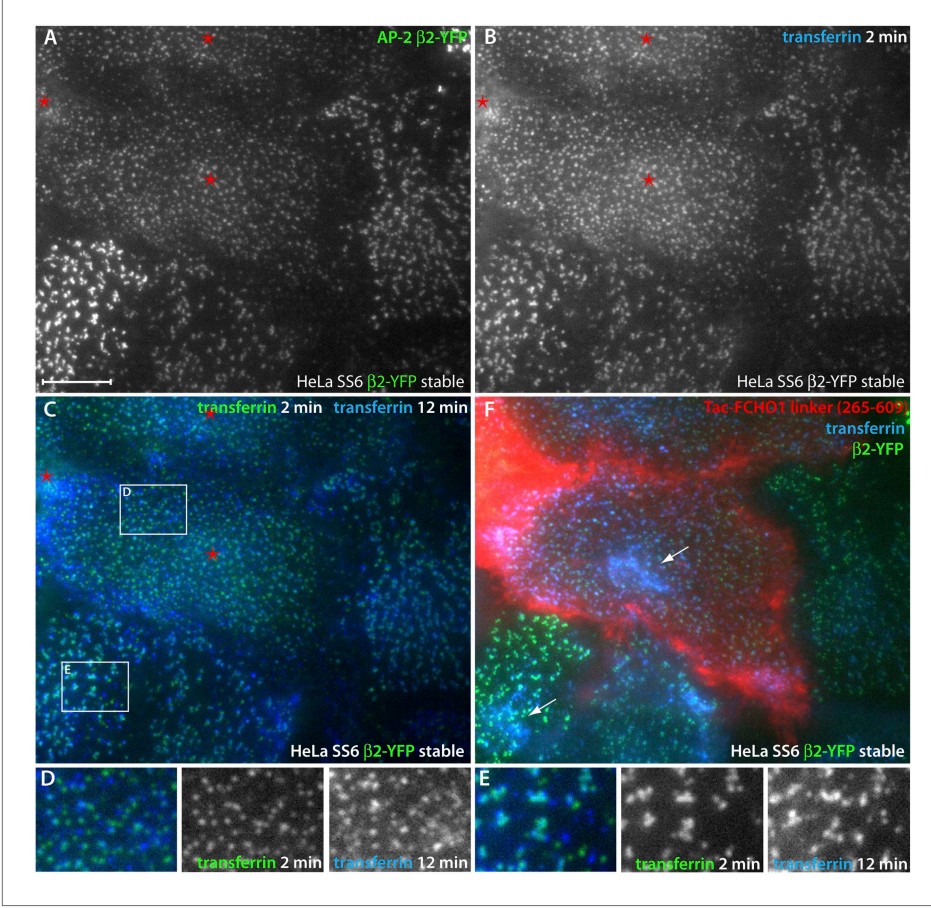

**Figure 14**. Direct regulation of coat morphology and cargo packaging by the FCHO1 linker domain. (**A** and **B**) Color channel separated total internal reflection images of HeLa SS6 cells stably expressing β2-YFP (**A**) and pulsed for 2 min with 25 µg/ml Alexa Fluor647 transferrin (**B**). The cells were previously transfected with Tac-FCHO1 linker (265–609) and Tac-expressing cells are indicated (asterisks). (**C**) Comparison of individual frames from the transferrin channel at 2 min and 12 min by pseudocoloring the initial fame green and overlaying the last frame colored blue. The Tac-expressing cells with the rearranged β2-YFP spots are indicated (asterisks). (**D** and **E**) Enlarged, color-channel separated views of the labeled boxed regions in (**C**). (**F**) After completion of time-resolved image acquisition, Alexa Fluor546-conjugated anti-Tac mAb 7 G7B6 was added to identify unambiguously the Tac-transfected cells. Following incubation with the anti-Tac, a final image was collected. Internalized transferrin within endosomes visible in the total internal reflection field is indicated (arrows). Scale bar: 10 µm.

DXF- and FXSXF-based AP-2-associating peptide sequences (*Meyerholz et al., 2005*), and can interact physically with clathrin (*Tebar et al., 1999*; *Meyerholz et al., 2005*). The protein also has two NPF tripeptide motifs that bind to EH domains (*Salcini et al., 1997*), as found in Eps15/R and intersectins. Yet CALM knock-out mice are born, and mouse embryonic fibroblasts can be cultured and analyzed (*Scotland et al., 2012*; *Suzuki et al., 2012*). In fact, a considerable number of clathrin coat components and ancillary proteins have been experimentally deleted or incapacitated without complete disruption of clathrin-dependent internalization (*Garcia et al., 2001*; *Kang-Decker et al., 2001*; *Morris et al., 2002*; *Kamikura and Cooper, 2003*; *Holmes et al., 2007*; *Koh et al., 2007*; *Wang et al., 2008*; *Chen et al., 2009*; *Mullen et al., 2012*; *Pozzi et al., 2012*; *Scotland et al., 2012*; *Suzuki et al., 2012*; *Umasankar et al., 2012*; *Kononenko et al., 2013*; *Tsushima et al., 2013*; *Alazami et al., 2014*). Although Fcho1 and Fcho2 were proposed to be all-important (*Henne et al., 2010*), we have carefully evaluated the overall veracity of this provocative idea. HeLa cells with undetectable levels of FCHO1 and FCHO2 still fabricate clathrin lattices and internalize clathrin-dependent transmembrane cargo. We have maintained these muniscin-depleted cells in culture for many months.

That is not to say that these proteins are insignificant. The cellular concentration of muniscins regulates clathrin-coated structure organization and dynamics and, thereby, the efficiency of cargo internalization. Clearly, this underscores the importance of these particular pioneers in clathrin lattice biogenesis and function. The enlarged clustered clathrin assemblies characteristic of essentially-munsicin-null HeLa cells appear to be linked to physiological compensation. Operationally, they resemble hot spots, areas of repeated vesicle budding from non-terminal clathrin puncta within a spatially restricted area (*Nunez et al., 2011*). Hot spots were actually reported the first time GFP-tagged clathrin-coated structures were visualized live (*Gaidarov et al., 1999*). Endocytic activity at hot spots is more resistant to perturbations in the levels AP-2 and PtdIns(4,5)P$_2$, so-called nucleation factors (*Nunez et al., 2011*). Thus, the extended long-lived lattices are likely better adapted to coupled budding events to conserve and reuse limiting factors and maintain steady, but slowed, clathrin-mediated endocytosis.

The reduced capacity to sequester transferrin (receptors) within the clathrin-coated structures that we note in clone 1.E cells could indicate that the unstructured linker domain of FCHO1/2 stabilizes the activated, open form of AP-2 to bias cargo capture. This idea is in line with the increased abundance of the smallest population of coated structures in the clone 1.E (and clone #64) cells, presumably abortive structures because of the overall reduced rate of transferrin uptake in these cells. However, live-cell imaging discloses that FCHO1/2 arrive at nascent clathrin bud sites on the cell surface with or before AP-2 (*Henne et al., 2010*; *Taylor et al., 2011*), and current copy number estimates suggest that AP-2 outnumbers FCHO2 in HeLa cells by an order of magnitude and FCHO1 by at least a factor of 1000 (*Kulak et al., 2014*). So the interaction between AP-2 and FCHO1/2 is unlikely to be stoichiometric within the lattice, as borne out by proteomic analysis (*Borner et al., 2012*). An obvious question, though, is why AP-2 is not able to properly engage the transferrin receptor YTFR sorting signal despite being positioned in the surface lattices of clone 1.E cells. As indicated above, it is well established that the heterotetramer displays multiple modes of attachment to membrane-apposed clathrin. In addition to four spatially discrete PtdIns(4,5)P$_2$ binding sites (*Gaidarov and Keen, 1999*; *Jackson et al., 2010*), AP-2 has at least two dedicated cargo-specific interactions surfaces (*Jackson et al., 2010*), a clathrin box (*Shih et al., 1995*) to contact the clathrin heavy chain terminal domain β-propeller (*ter Haar et al., 2000*), as well as a second site in the β2 subunit appendage (*Edeling et al., 2006*) that binds to the extended α-solenoid region of the clathrin heavy chain designated the ankle (*Knuehl et al., 2006*). Both the α- and β2-subunit appendages interact with a large overlapping set of early-arriving clathrin-associated sorting and accessory proteins. The appendages do not depend on allosteric changes within the core to promote binding because AP-2 can be quantitatively affinity purified with immobilized appendage binding motifs or immunoprecipitated with mAb AP.6 (*Chin et al., 1989*), which binds to the platform interaction site of the α appendage. We suspect that the wide range of interaction possibilities allows deposition of some incompletely rearranged AP-2 at clathrin lattices formed in the absence of muniscins. AP-2 can transition between metastable states; a partially open form bound to the dileucine sorting signal has been characterized (*Kelly et al., 2008*). An incomplete conformational transition in all membrane-associated AP-2 in the clone 1.E cells may also be linked to the alterations in lattice arrangement we see. Using synthetic liposome templates, AP-2 driven clathrin-coated buds are spherical (*Kelly et al., 2014*) while clathrin coats nucleated by a monomeric clathrin binding domain are flat but stabilize membrane remodeling induced by thermal fluctuations (*Dannhauser and Ungewickell, 2012*).

The crescent-shaped EFC domain in the FCHO proteins prompted the conjecture that initial dimpling of a spherical membrane patch defined a bud site (*Henne et al., 2010*). Further, if the munscins, and their pioneer binding partners collectively assembled into a circumferential rim defining the outer edge of the indented membrane, they could stabilize nascent buds possibly by preventing lateral diffusional escape of phosphoinositides (*Zhao et al., 2013*) necessary for successful placement of incoming inner layer coat constituents. Yet two lines of evidence argue against this notion: First, the linker region alone is able to correct or remodel the abnormal distribution of clathrin structures, even when not directly membrane associated. Second, the Tac-linker + μHD chimera, while membrane spanning, is unlikely to similarly remodel membrane topology adjacent to the linker, but it is still able to restore the organization of surface clathrin in clone #64/1.E cells. Instead, the unstructured linker tract between the folded EFC and μHD domains in FCHO1, and the sequence-related portion of Sgip1, are able to drive membrane deposition of AP-2 and partner engagement. In this sense, muniscins could operate in a conceptually analogous manner to the small GTPase Arf1 activating the structurally and functionally related heterotetrameric AP-1 clathrin adaptor (*Ren et al., 2013*).

Our findings allow us to rationalize a number of recent studies with an integrated model for the biogenesis of clathrin lattices at the cell surface. We concur that both AP-2 and Fcho1/2 are central regulators during clathrin coat initiation; both are clearly core members of the pioneer module of endocytic components (*Henne et al., 2010*; *Taylor et al., 2011*; *Cocucci et al., 2012*). A pivotal assembly step involves the conformational rearrangement of AP-2 (*Collins et al., 2002*; *Höning et al., 2005*; *Kelly et al., 2008*; *Jackson et al., 2010*; *Cocucci et al., 2012*; *Kelly et al., 2014*). This shift in the equilibrium from the closed cytosolic basal conformation to the open, assembly-competent form of AP-2 can be regulated synergistically by PtdIns(4,5)P$_2$ and tyrosine (YXXØ)- or dileucine ([DE] XXXL[LIM]-based internalization signals). Reorganization of AP-2 by these allosteric regulators could account for the reported minimal stoichiometry of two AP-2 heterotetramers and one or two clathrin trimers to nucleate a clathrin-coated structure (*Cocucci et al., 2012*). Indeed targeted gene disruption of the AP-2 µ2 subunit is pre-implantation lethal in mice (*Mitsunari et al., 2005*). But in *C. elegans* and *S. cerevisae*, AP-2 is dispensable (*Huang et al., 1999*; *Yeung et al., 1999*; *Mayers et al., 2013*). In yeast, seven early module (pioneer) proteins can be deleted without fully disrupting clathrin-dependent internalization (*Brach et al., 2014*). As noted above, loss of numerous mammalian pioneer factors (Eps15, CALM, Hrb, epsin, Dab2, intersectin, Necap 1) similarly does not terminate clathrin-mediated endocytosis. One interpretation of these results is that there are several parallel pathways to arrive at an internalization competent clathrin assemblage at the plasma membrane. However, extinguishing the muniscins has a more severe effect than loss of a set of pioneers that physically engage the µHD. This is plainly apparent in cells expressing the tailored Tac-µHD protein—despite massive misrouting of EPS15 (and EPS15R and intersectin) to the juxtanuclear region, AP-2-positive puncta at the cell surface persist. Our findings indicate that FCHO1, and specifically the linker segment, interacts with AP-2 and can promote membrane deposition of the heterotetramer in the absence of elevated levels of PtdIns(4,5)P$_2$. Further, because AP-2 cannot bind productively to clathrin in the closed basal conformation (*Kelly et al., 2014*), our biochemical experiments showing that FCHO1 can modulate the clathrin binding properties of EPS15-bound AP-2 suggest that FCHO1 can provide an alternative pathway to rearrange AP-2 to promote cargo capture, clathrin lattice assembly and budding. The remarkable effect that a membrane-attached FCHO1 linker has on the disposition of surface clathrin structures supports this idea. The appearance of expanded planar clathrin sheets, which often contain assembly defects in cells depleted of muniscins (FCHO2 principally), argues additionally for an important role of these proteins in overseeing swift and regular polymerization of the clathrin coat. Necap 1 also exerts a regulatory input on AP-2 by competitively engaging several binding surfaces on the heterotetramer (*Ritter et al., 2013*). As FCHO1 and Necap 1 associate with each other, whether the effect of the FCHO1 linker directly involves Necap 1 will be important to establish. Finally, our results defining regulation of overall size of clathrin-coated buds raise the possibility that kinetic uncoupling between the outer and inner layer components of other vesicular coats could also alter the morphology of the forming tubulovesiclular carrier to accommodate variably sized cargo.

## Materials and methods

### Construction of TALEN gene-edited cells

All TALENs were designed using open-access software (Mojo Hand; http://www.talendesign.org) as described (*Bedell et al., 2012*). The repeat variable di-peptide (RVD)-containing units for *FCHO2* TAL-1 (NG NN NN NI NG HD NG NG NN NG NG NI NN NI NI NI) and *FCHO2* TAL-2 (NI HD NG NG HD NG NN NI NI HD NG NG HD HD NG NG) or *FCHO1* TAL-1 (HD HD NG NN NG NI HD HD NI HD NI NN HD NN NG NN NI) and *FCHO1* TAL-2 (HD HD NN HD HD NI NN HD NG HD HD NG NG NN NN NG NN) were assembled using the Golden Gate approach (*Cermak et al., 2011*). After assembly, the RVDs were cloned into the pC-Goldy TALEN destination vector (*Bedell et al., 2012*; *Carlson et al., 2012*) suitable for expression in mammalian cells (Addgene, Boston, MA).

For TALEN transfections, equal amounts of each pair of TALENs were introduced into a HeLa cells grown in a 35 × 10 mm tissue culture dish using Lipofectamine 2000 (Invitrogen/Life Technologies, Grand Island, NY). Transfected cells were cultured 3 days at 30°C, before splitting for indel analysis and plating for clone selection. Individual clones were collected by a standard limiting dilution approach in 96-well plates. Clonal populations were expanded and screened by SDS-PAGE and immunoblotting with antibodies specific for human FCHO2. To confirm gene disruption in FCHO1 mutants, genomic DNA was isolated from individual clonal populations or from the transfected-cell pool using DNeasy

Blood and Tissue kit (Qiagen, Valencia, CA). The genomic region surrounding the target site was PCR amplified with human *FCHO1* locus specific primers using the genomic DNA template obtained from each clone. PCR amplicons were initially digested with appropriate restriction enzymes and the mutations were assessed by loss of restriction enzyme digestion. Then the poly-allelic mixture of the individual clones was ligated to TOPO-TA vector (Invitrogen) and analyzed by sequencing.

## Molecular cloning and constructs

GFP-FCHO1 was designed as described previously (*Reider et al., 2009*; *Umasankar et al., 2012*). To generate GFP-FCHO2, full length human FCHO2 (1–810) was PCR amplified from cDNA clone #9021067 (Open Biosystems, Lafayette, CO) and inserted into pEGFP-C1 using SalI restriction sites and Cold Fusion cloning technology (System Biosciences, Mountain View, CA). Similarly, GFP-Sgip1 was constructed by Cold Fusion cloning of full-length mouse Sgip1 (1–806) from cDNA clone #4482298 (Open Biosystems) into pEGFP-C1 using BglII restriction sites. GFP-FCHO1 (265–889), GFP-FCHO1 (265–609) and GFP-FCHO1 (609–889) were described previously (*Umasankar et al., 2012*). The various N-terminally-tagged truncation constructs used for the functional complementation studies were obtained by introducing stop codons at appropriate sites using QuikChange site-directed mutagenesis (Stratagene/Agilent Technologies, Santa Clara, CA). The deletion constructs GFP-FCHO1 (1–889; Δ316–467 or Δ316–339) or GST-eps15 (595–896; Δ617–636) were produced using Phusion site-directed mutagenesis (Thermo Scientific, Pittsburgh, PA).

The muniscin linker regions fused to an N-terminal GST (GST-FCHO1 (316–467), GST-FCHO2 (314–444) and GST-Sgip1 (77–214) plasmids) were generated by the insertion of appropriate PCR amplicons into EcoRI site in pGEX-4T-1 by Cold Fusion cloning. The GST-β2 appendage (rat residues 701–937), GST-$\alpha_C$ appendage (mouse residues 701–938), GST-EPS15 (595–896) and GST-ARH constructs have been described previously. The plasmid encoding the GST-PHear domain of mouse Necap 1 was kindly provided by Dr Brigitte Ritter.

Tac in the pcDNA3.1 plasmid is explained elsewhere (*Jha et al., 2012*). FCHO1 μHD (residues 609–889) or linker + μHD (residues 265–889) were PCR amplified from GFP-FCHO1 and cloned into the Tac plasmid between EcoRV and NotI restriction sites to replace the endogenous Tac cytosolic segment. A stop codon at residue 610 in Tac-FCHO1 linker + μHD yielded Tac-FCHO1 linker (265–609). All constructs were verified by automated dideoxynucleotide sequencing (Genewiz, South Plainfield, NJ), and the primer and restriction site details, sequences (*Supplementary file 1*), and maps are all available upon request.

## Cell culture, transfections and immunofluorescence

HeLa SS6 (*Elbashir et al., 2001*), HeLa clone #64, clone #64/1.E, the neuronal SH-SY5Y (*Biedler et al., 1978*) and MCF-7 (*Soule et al., 1973*) cells were cultured in DMEM supplemented with 10% fetal calf serum and 2 mM L-glutamine at 37°C in an atmosphere of 5% $CO_2$. HeLa SS6 β2-YFP stably expressing cells (*Keyel et al., 2008*) were grown in the same medium containing 0.5 mg/ml G418. K562 cells (*Lozzio and Lozzio, 1975*) were grown in suspension in RPMI media supplemented with 5% fetal calf serum and 2 mM L-glutamine at 37°C in 5% $CO_2$.

Cells were transfected with plasmids using Lipofectamine 2000 (Invitrogen) or with siRNA oligonucleotides using Oligofectamine (Invitrogen) according to the manufacturer's recommendations (*Umasankar et al., 2012*). After 18–24 hr, cells were fixed with 4% paraformaldehyde in PBS (pH 8.0) and quenched and permeabilized with a mixture of 75 mM $NH_4Cl$, 20 mM glycine and 0.1% Triton X-100. The cells were blocked in 5% normal goat serum diluted in PBS/fish skin gelatin/saponin mixture and stained with various antibodies. For fluorescent transferrin binding and uptake assays, cells were preincubated in DMEM, 25 mM Hepes, 0.5% BSA at 37°C for 1 hr to remove bound transferrin.

## Reverse transcriptase-polymerase chain reaction (RT-PCR)

RNA was isolated from HeLa SS6 and SH-SY5Y cells by dissolving pelleted cells directly in TRIzol (Invitrogen) followed by chloroform extraction. The aqueous layer was precipitated with isopropanol and resuspended in DEPC water. To generate cDNA, the SuperScript III kit (Invitrogen) was used with 4 μg DNase-treated RNA according to the manufacturer's instructions. PCR of endocytic protein transcripts was performed with Taq polymerase (Genscript, Piscataway, NJ) and the following primers: *FCHO1* sense 5′-CTG GCG CTG TGC CAC CTG GAA CT-3′, *FCHO1* antisense 5′-GTA CTC TCC CTC CGC AGC CGC TC-3′, *FCHO2* sense 5′-CCC CAG CAA TAT CTA GAC ACA GTC C-3′, *FCHO2* antisense 5′-TAC AGA AAG AGG AGT TGT GGG CC-3′, *SGIP1* sense 5′-GAA GTG GCA AGA CCC AGG CGT TCC-3′, *SGIP1* antisense 5′-GGA GGT GTT CCA GTG GGA AAA GGC C-3′, β-actin sense 5′-GAG AGG CAT CCT CAC CCT GAA GTA C-3′, β-actin antisense 5′-GCA CAG CCT GGA TAG CAA CGT ACA T-3′,

clathrin heavy chain sense 5′-GGA AGG AGA GTC TCA GCC AGT GAA A-3′, clathrin heavy chain anti-sense 5′-TAT GTA ACT TCC CTC CAG CTT GGC C-3′, *EPS15* sense 5′-TTG TTG CAG CAA GCG ATT CAG CCA-3′, *EPS15* antisense 5′-AGG GCA GGG TCT TGT TGG AGT TCC-3′, *EPS15R* sense 5′-AGC CTC AAC AGC ACA GGG AGC CTG-3′, *EPS15R* antisense 5′-AAG GTT CTG GGT GAG GCC CGA GTG-3′. The PCR cycling conditions used were: 94°C for 3 min, then 94°C for 30 s, 53°C (*FCHO2* and clathrin) or 55°C (remainder) for 30 s and 72°C for 1 min, repeated for 30 cycles, and then final extension at 72°C for 7 min.

## Antibodies

The affinity-purified rabbit anti-FCHO1 antibody 1 (1:2500), anti-FCHO2 (1:2500), anti-Dab2 (1:1000), anti-epsin 1 (1:10000) and anti-AP-1/2 β1/β2-subunit GD/1 (1:2500) antibodies were produced for our laboratory. A second affinity-purified anti-FCHO2 (1:2500) antibody was kindly provided by Dr Harvey McMahon. Affinity-purified rabbit anti-eps15 polyclonal antibody and anti-AP-1/2 β1/β2-subunit mAb 100/1 were gifts from Dr Ernst Ungewickell, the anti-clathrin HC mAbs TD.1 (1:5000) and X22 and the anti-AP-2 α-subunit mAb AP.6 were generously provided by Dr Frances Brodsky, the rabbit R11-29 anti-AP-2 μ2-subunit (1:3000) antiserum kindly provided by Dr Juan Bonifacino, and the rabbit anti-intersectin 1 (1:2000) and rabbit anti-NECAP 1 (1:5000) antibodies were a gift from Dr Peter McPherson. The goat anti-Hrb C-19 (1:1000; sc-1424), rabbit anti-eps15 C-20 (1:500; sc-534) and goat anti-CALM C-18 (1:500; sc-6433) polyclonal antibodies and rabbit anti-SHIP2 mAb E−2 (1:500; sc-166641) from Santa Cruz Biotechnology (Dallas, TX), rabbit anti-EPS15R (1:5.000; EP-1145Y) antibody and rabbit anti-FCHO1 antibody 2 (1:1000; ab84740) from AbCAM (Cambridge, MA) and the mouse anti-myc mAb 9E10 (1: 1000; MMS-150P) from Biolegend (Dedham, MA) were used. The anti-β tubulin mAb E7 (1:2500) was purchased from the Developmental Studies Hybridoma Bank. The mAbs directed against the AP-2 α subunit clone 8/Adaptin α (1:1000; 610502) and μ2 subunit clone 31/AP50 (1:500; 611350) were from BD Transduction Laboratories (San Jose, CA). Another AP-2 α subunit mAb C-8 (1:500; sc-17771) came from Santa Cruz and the anti-Tac mAb 7 G7B6 was from Ancell (Bayport, MN). The secondary antibodies used were donkey anti-rabbit (1:5000; NA934V)- or anti-mouse (1:5000; NA931V)- horseradish peroxidase conjugates from GE Healthcare Life Sciences (Pittsburgh, PA) or rabbit anti-goat (1:5000; A4174) peroxidase conjugate from Sigma (St. Louis, MO).

## GST pull-down assays and immunoblotting

The purification of GST, various GST-fusion proteins, and the preparation of rat brain cytosol for GST-based pull-down assays have been thoroughly described previously (*Umasankar et al., 2012*). The lysates from parental HeLa SS6, HeLa clone #64 or HeLa clone #64/1.E cells were prepared from cells detached with Cellstripper (Cellgro/Mediatech, Manassas, VA). K562 cells grown in suspension were pelleted directly. After washing, cell pellets were solubilized on ice for 30 min in 25 mM Hepes-KOH pH 7.2, 125 mM potassium acetate, 5 mM magnesium acetate, 2 mM EDTA, 2 mM EGTA, and 2 mM dithiothreitol (assay buffer) supplemented with 1% Triton X-100, 1 mM PMSF and complete protease inhibitor cocktail (Roche, Indianapolis, IN). Lysates were centrifuged at 20,000×*g* before use in binding assays. Assays were in assay buffer, usually in a final volume of 300 μl. After incubation at 4°C for 1–2 hr, glutathione-Sepharose beads were sedimented and washed four times with ice-cold PBS. Aliquots of the supernatant and washed pellet factions were resolved by SDS-PAGE and either stained with Coomassie blue or transferred to nitrocellulose for immunoblotting. Detection was with enhanced chemiluminescence and X-ray film. The hetero-tetrameric core of AP-2 was produced in *Escherichia coli* BL21 (DE3) pLysS after transfection with two bicistronic plasmids encoding the β2-subunit trunk and myc-tagged μ2 subunits (Amp$^r$) and the GST-α-subunit trunk and σ2 subunit (Kan$^r$) generously provided by David Owen. Purification was exactly as described (*Collins et al., 2002*; *Kelly et al., 2008*; *Jackson et al., 2010*). For the binary interaction assays, a modified assay buffer composed of 25 mM Hepes-KOH pH 7.2, 25 mM Tris–HCl, 125 mM potassium acetate, 5 mM magnesium acetate, 2 mM EDTA, 2 mM EGTA, 10 mM dithiothreitol, 0.2% Igepal CA-630 (Nonidet P40 substitute) and 0.1 mg/ml BSA was used. After incubation of immobilized GST-fusion proteins with 20 μg/ml purified AP-2 core in modified assay buffer for 60 min at 4°C, the glutathione-Sepharose beads were recovered at 1000 × *g* for 1 min. Following a first wash in ice-cold modified assay buffer and centrifugation at 1000×*g* for 1 min, the beads were washed two additional times with ice-cold PBS and centrifugation at 10,000×*g* for 1 min.

## Microscopy and live-cell imaging

Confocal fluorescence images were collected on an Olympus Fluoview FV1000 microscope as described previously (*Keyel et al., 2006*; *Umasankar et al., 2012*). The z-stacks were collected with a 0.25-μm step size between optical sections and the stacks were deconvolved using the blind

deconvolution algorithm within Autoquant X3 (Media Cybernetics, Rockville, MD). Quantitation of objects in fluorescent images was with the Nikon Elements software (version 4.30, Nikon, Melville, NY).

For three-color total internal reflection fluorescence microscopy, adherent cells were imaged on a Nikon Eclipse Ti inverted microscope with a 60 × 1.49 NA oil-immersion objective. Cells were maintained in DMEM supplemented with 10% fetal calf serum and 25 mM HEPES, pH 7.2 at 37°C on MatTek dishes (MatTek Corporation, Ashland, MA) and imaged continuously at 5 s/frame. GFP/YFP was excited with a 488 nm laser, Alexa Fluor546 conjugated anti-Tac mAb with a 561 nm laser, and transferrin-Alexa 647 (Molecular Probes/Life Technologies, Grand Island, NY) with a 647 nm laser line. Images were collected using an Andor (Belfast, Ireland) Xyla 5.5 camera; at full resolution under these conditions the pixel size with a 1 × coupler matches Nyquist sampling (120 nm *xy* exactly). Data sets were acquired acquired using Nikon Elements.

## Electron microscopy

Adherent plasma membrane from parental HeLa SS6 and clone #64/1.E cells were prepared for rapid-freeze, deep-etch electron microscopy as described previously (*Heuser, 2000*). Briefly, cells grown on oriented pieces of glass cover slip were disrupted by sonication and fixed in a glutaraldehyde and paraformaldehyde mixture before flash freezing in liquid helium (*Keyel et al., 2006*).

## Acknowledgements

We are grateful to Souvik Chakraborty for his contribution to the early stages of this work, to Jenny Minyoung Park for technical assistance, and to Robyn Roth for her enormous skill and dedication in generating the deep-etch EM data. We are extremely grateful to Juan Bonifacino, Frances Brodsky, Adam Linstedt, David Owen, Peter McPherson, Brigitte Ritter, and Ernst Ungewickell for graciously providing important reagents that made these studies possible. This work was supported by NIH R01 GM106963 to LMT.

## Additional information

### Funding

| Funder | Grant reference number | Author |
| --- | --- | --- |
| National Institute of General Medical Sciences | R01 GM106963 | Linton M Traub |

The funder had no role in study design, data collection and interpretation, or the decision to submit the work for publication.

### Author contributions

PKU, LMT, Conception and design, Acquisition of data, Analysis and interpretation of data, Drafting or revising the article; LM, JRT, Conception and design, Acquisition of data, Analysis and interpretation of data; AJ, SCW, Acquisition of data, Analysis and interpretation of data; BD, Acquisition of data, Contributed unpublished essential data or reagents

## Additional files

### Supplementary file

• Supplementary file 1. List of various constructs used in this study and the sets of specific primers, restriction sites, plasmids and the methods of cloning used to design these constructs.

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
