## [Decision Letter]

Thank you for sending your work entitled “A clathrin coat assembly role for the muniscin protein central linker revealed by TALEN-mediated gene editing” for consideration at *eLife*. Your article has been favorably evaluated by Randy Schekman (Senior editor) and 3 reviewers, one of whom, Suzanne Pfeffer, is a member of our Board of Reviewing Editors.

The Reviewing editor and the other reviewers discussed their comments before we reached this decision, and the Reviewing editor has assembled the following comments to help you prepare a revised submission.

There is tremendous interest in the earliest events underlying clathrin mediated endocytosis, and this careful study explores the roles of FCHO 1 and 2, previously implicated as being key and essential for endocytosis initiation. The authors created a HeLa cell in which FCHO1 and FCHO2 genes are targeted using TALEN endonucleases, in an effort to clarify confusion that has resulted from other labs achieving different extents of siRNA depletion. They report that a central FCHO1 linker region is key to AP2 recruitment.

The experiments are carried out with great care yet the paper would benefit from some tightening and clarification of the figures to guide non-experts through the comparisons. The reviewers have two major requests.

First, it would be great to show direct interaction of AP2 with the linker region. This would make the story very complete. You showed previously that alpha and beta2 appendages both bind to FCHO1; it should be very easy to test if binding occurs via the linker.

Second, you conclude that these proteins “impact the fidelity of the polyhedral construction” and are not “obligatory for sustained clathrin lattice growth”. These conclusions would be bolstered significantly by at least some quantitation of marker protein distributions in fluorescent micrographs (size of stained objects from Cellprofiler or ImageJ for quantitative comparison) and careful assessment of the EM images as follows:

In Figure 6, (Please show magnification bars in all figures) the authors conclude that clone 1.E cells “display geometrically poorly ordered lattices” yet this is hard to discern from the images shown. Please quantify the shapes (# and size of pentagons/hexagons/heptagons), diameters of patches, etc. to provide metrics that support the conclusions. The use of deep etch to score this phenotype is a really nice part of the paper, but the differences need to be made more clear.

In summary, an excellent and interesting contribution.

---

## [Author Response]

*The experiments are carried out with great care yet the paper would benefit from some tightening and clarification of the figures to guide non-experts through the comparisons. The reviewers have two requests*.

*First, it would be great to show direct interaction of AP2 with the linker region. This would make the story very complete. You showed previously that alpha and beta2 appendages both bind to FCHO1; it should be very easy to test if binding occurs via the linker*.

We agree that this issue is central to our claim that the FCHO1/2 linker polypeptide can act as a regulator of AP-2 conformation and membrane association. To address whether we could detect a physical interaction between the FCHO1/2 linker and AP-2, we performed two additional sets of experiments.

The first was to interrogate whether the a appendage of AP-2 binds to the FCHO1 linker region, because as the reviewers correctly state, we published previously that the a appendage (and to a lesser extent the b2 appendage) can pull down FCHO1 in biochemical assays. For this we used the competition pull-down format. Soluble AP-2 in rat brain cytosol binds to bead-bound GST-FCHO1 linker (316-467), but addition of a large excess of the AP-2 a appendage has no appreciable effect on this binding. Under the same conditions, the C-terminal tract of EPS15 fused to GST also binds AP-2 from brain cytosol but added a appendage clearly competes with the intact heterotetramer in a dose-dependent manner. We conclude from this work that it is NOT the AP-2 appendages that bind to the FCHO1 unstructured linker sequence.

More important, we then conducted protein interaction assays with the purified recombinant core of AP-2. This heterotetrameric complex binds physically to the FCHO1, FCHO2 and Sgip1 linker regions, and the presence of the bound AP-2 subunit chains is confirmed by immunoblotting. We conclude that the linker polypeptide engages the AP-2 core complex directly, which is certainly more gratifying and mechanistically understandable if this interaction operates by changing the conformation of AP-2 to promote membrane attachment, cargo capture and coat assembly.

These new results are by far the most significant addition to the revised version, and the accompanying narrative text is now added to the manuscript as part of the extended Figure 9 and supplement. We are very grateful to the reviewers for urging us to extend our work and ascertain that the linker interaction is both direct and involves the tetrameric core of AP-2.

*Second, you conclude that these proteins “impact the fidelity of the polyhedral construction” and are not ”obligatory for sustained clathrin lattice growth”*. *These conclusions would be bolstered significantly by at least some quantitation of marker protein distributions in fluorescent micrographs (size of stained objects from Cellprofiler or ImageJ for quantitative comparison) and careful assessment of the EM images as follows:*

*In*
Figure 6*, (Please show magnification bars in all figures) the authors conclude that clone 1.E cells “display geometrically poorly ordered lattices” yet this is hard to discern from the images shown. Please quantify the shapes (# and size of pentagons/hexagons/heptagons), diameters of patches, etc. to provide metrics that support the conclusions. The use of deep etch to score this phenotype is a really nice part of the paper, but the differences need to be made more clear*.

As requested, we have added scale bars to all of the morphologic images. Further, we have used Nikon Elements software package to measure size distribution parameters and now state the findings in the text:

“Comparative quantitative analysis of the distribution of AP-2 in HeLa SS6 and the clone 1.E cells indicates a sharp increase in the very smallest (<60 nm^2^) coated structures on the ventral surface. In the parental cells, these account for <25% of the AP-2 signal while in the FCHO1/2-depleted cells these small puncta represent almost 45% of the total AP-2. There is also an increase in the largest (>600 nm^2^) clathrin assemblages, as previously seen in FCHO2 silenced HeLa cells (79; 113). After disruption of the FCHO1 and FCHO2 genes, 46% of membrane-associated AP-2 is present in structures >600 nm^2^ while only 29% of AP-2 occurs within this size category in the HeLa SS6 cells. Analogous results are obtained upon measuring the distribution of EPS15 in the two cell types. Engineered loss of muniscins therefore shifts the equilibrium of clathrin lattices at the cell surface to the very small and the largest forms.”

For the quantification of the freeze-etch EM micrographs, we attempted to provide metrics as urged by the reviewers. Unfortunately, the difficulty is that, as we are trying to convey, in our preparations of replicas from the FCHO1/2 doubly depleted HeLa clone #64/1.E cells, the ultrastructure of the lattices is generally too distorted to accurately count the full number of hexagons, pentagons and heptagons. This is illustrated by an enlargement from the clone #64/1.E cell freeze-etch data sets, shown for the reviewers below:Author response image 1.

There are numerous instances where it is difficult to define the precise edges of a polygon within the array. We are unable to provide an accurate metric at this time; this will be the subject of additional future analysis. In any event, what we are basically trying to display with these images is that there are discernable ultrastructural differences between the parental planar clathrin lattices and those that are present in the deep-etch replicas of clone #64/1.E cells. We have now prepared a second rapid freeze-deep etch image set (Figure 6—figure supplement 1) that we believe answers qualitatively the reviewers query about the difficulty in discerning the geometric differences from the images previously shown. In the new figure, enlargements from two similarly sized zones of clathrin in either the HeLa SS6 or HeLa clone #64/1.E cells are presented. The regular features of the normal parental arrays are highlighted, and the abnormalities in the clone #64/1.E cells are identified. The fact that there are planar clathrin assemblies in the clone #64/1.E cells forms basis for our assertion that FCHO1/2 are not “obligatory for sustained clathrin lattice growth”.

Ultimately, this is not a central point of the manuscript, but the number of separate images we now show should make it clear that the abnormal lattices are reproducibly seen in our data set. With the current time constraints, we feel that a thorough morphological description and quantitative treatment of the lattice changes is out of the scope of the present study. Yet, given the entire collection of results presented in this manuscript, we hope that the reviewers will be satisfied with the supplement that better delineates (albeit qualitatively) the ultrastructural abnormalities in the FCHO1/2-depleted cells.